# Low-Rank Head Avatar Personalization with Registers

**Sai Tanmay Reddy Chakkera**
Department of Computer Science
Stony Brook University
schakkera@cs.stonybrook.edu

**Aggelina Chatziagapi**
Department of Computer Science
Stony Brook University
echatziagapi@cs.stonybrook.edu

**Md Moniruzzaman**
Atmanity Inc.
mman@atmanity.io

**Chen-Ping Yu**
Atmanity Inc.
cpyu@atmanity.io

**Yi-Hsuan Tsai**
Atmanity Inc.
yhtsai@atmanity.io

**Dimitris Samaras**
Department of Computer Science
Stony Brook University
samaras@cs.stonybrook.edu

## Abstract

We introduce a novel method for low-rank personalization of a generic model for head avatar generation. Prior work proposes generic models that achieve high-quality face animation by leveraging large-scale datasets of multiple identities. However, such generic models usually fail to synthesize unique identity-specific details, since they learn a general domain prior. To adapt to specific subjects, we find that it is still challenging to capture high-frequency facial details via popular solutions like low-rank adaptation (LoRA). This motivates us to propose a specific architecture, a Register Module, that enhances the performance of LoRA, while requiring only a small number of parameters to adapt to an unseen identity. Our module is applied to intermediate features of a pre-trained model, storing and re-purposing information in a learnable 3D feature space. To demonstrate the efficacy of our personalization method, we collect a dataset of talking videos of individuals with distinctive facial details, such as wrinkles and tattoos. Our approach faithfully captures unseen faces, outperforming existing methods quantitatively and qualitatively. Project page: https://starc52.github.io/publications/LoRAvatar/.

## 1 Introduction

Synthesizing photo-realistic human faces has long been a challenge for both computer vision and graphics. It has broad applications from AR/VR, virtual communication, and video games, to the movie industry and healthcare. Earlier approaches rely on 3D morphable models (3DMMs) (Garrido et al., 2015, 2014; Thies et al., 2016), while subsequent methods turn to generative adversarial networks (GANs) (Kim et al., 2018; Pumarola et al., 2020; Prajwal et al., 2020; Vougioukas et al., 2020). More recent works learn 3D neural representations of the human face, which rely on neural radiance fields (NeRFs) (Pumarola et al., 2021; Park et al., 2021a; Gafni et al., 2020; Park et al., 2021b) or 3D Gaussian Splatting (3DGS) (Kerbl et al., 2023; Cho et al., 2024; Qian et al., 2024; Xu et al., 2024c). While these approaches lead to high-quality results, they usually require identity-specific training and are not able to generalize. Only a few recent methods propose *generic* models,

39th Conference on Neural Information Processing Systems (NeurIPS 2025).

e.g., GAGAvatar (Chu and Harada, 2024), which preserve the high-quality rendering of 3DGS, while trained on a large-scale dataset of multiple identities, enabling generalization to unseen human faces.

However, such generic models usually fail to produce key identity-specific facial details, since they learn a general domain prior. To produce distinctive details, prior work proposes adapting a pre-trained model to a specific identity, e.g. through fine-tuning or meta-learning (Nitzan et al., 2022; Zhang et al., 2023a; Saunders and Namboodiri, 2024). Low-rank adaptation (LoRA) (Hu et al., 2022) has been first proposed for large language models (LLMs). It injects trainable rank decomposition matrices into each layer of a pre-trained model, leading to a significant decrease of the learnable parameters and on-par performance compared to fine-tuning the entire model.

In this work, we address the problem of adaptation, also called *personalization*, to a specific identity, which is not seen in the initial training of a generic model for head avatar generation. Due to its efficiency and popularity in other fields, we start with LoRA, by learning low-rank decomposition matrices for specific layers. We notice that LoRA is not sufficient to synthesize high-frequency facial characteristics (see Figure 1). Inspired by Darcet et al. (2023) that learn additional tokens (registers) in order to store global information for a transformer network, we propose a specific module that extends the idea of registers to 3D registers for human faces. To the best of our knowledge, we believe that this is the first method to extend registers to 3D representations.

More specifically, we design a Register Module that learns a 3D feature space that stores and repurposes information for a human face during training. Similar to registers in ViT (Darcet et al., 2023) that store global information of an image, our Register Module stores the distinctive details of an identity, given different views. We apply our Register Module to intermediate features that are extracted from a pre-trained DINOv2 model (Oquab et al., 2023). While our proposed module can be applied to any network that uses DINOv2 features, we focus our study on GAGAvatar (Chu and Harada, 2024) as our generic pre-trained model since it is a highly competitive method trained on a large-scale dataset (VFHQ Xie et al. (2022)) and achieves state-of-the-art results in the general avatar domain. To evaluate the efficacy of our low-rank personalization, we collect a dataset of talking videos of individuals with rare high-frequency facial details, such as wrinkles and tattoos, that are not included in existing datasets. Our method outperforms state-of-the-art approaches, like meta-learning and vanilla LoRA, both quantitatively and qualitatively, while it only requires a small number of parameters to adapt.

In brief, our main contributions are as follows:

- We propose a novel method for low-rank personalization of a generic model for head avatar generation, that captures identity-specific facial details.

- We design a Register Module that stores and repurposes information for an identity in a learnable 3D feature space, extending the idea of registers for ViTs to 3D human faces.

- We collect a dataset, namely RareFace-50, of talking videos of individuals with distinctive facial characteristics, e.g. wrinkles and tattoos, that are challenging to synthesize with generic models, and thus demonstrating the need for our method.

## 2 Related Work

**Human Portrait Synthesis.** Earlier approaches for video synthesis of human faces are based on 3DMMs (Garrido et al., 2015, 2014; Thies et al., 2016). A 3DMM (Blanz and Vetter, 1999) is a parametric model that represents a face as a linear combination of the principal axes of shape, texture, and expression, learned by principal component analysis (PCA). Subsequent works propose GAN-based networks for video synthesis (Kim et al., 2018; Siarohin et al., 2019; Pumarola et al., 2020; Roich et al., 2023; Tancik et al., 2021; Buehler et al., 2024) and audio-driven talking faces (Prajwal et al., 2020; Zhou et al., 2021; Vougioukas et al., 2020; Xu et al., 2024a). GANs are usually trained on large datasets of 2D videos of multiple identities, but they cannot model the 3D face geometry. More recent works learn 3D neural representations of the human face, which rely on neural radiance fields (NeRFs) (Mildenhall et al., 2020) or 3D Gaussian Splatting (3DGS) (Kerbl et al., 2023). Diffusion models have also become popular but they only produce 2D videos (Xu et al., 2024b) or are identity-specific (Kirschstein et al., 2024a). In this paper, we explore personalization to capture identity-specific facial details by adapting a generic avatar model.

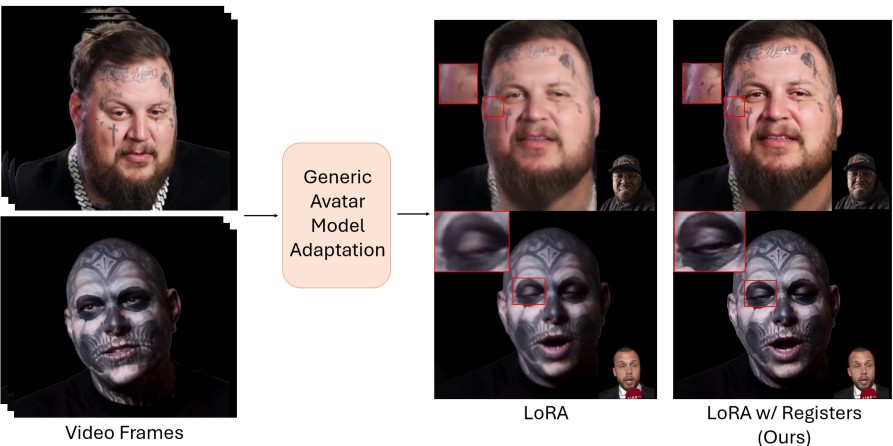

Video Frames     LoRA     LoRA w/ Registers (Ours)

Figure 1: Our method personalizes and adapts a generic head avatar model using LoRA, while preserving high-frequency identity-specific facial details using our Register Module and retaining the original inference speed. Note that the small image in the bottom-right corner is the driving image.

**Animatable 3D Head Avatars.** NeRFs have been first proposed for novel-view synthesis of static scenes (Mildenhall et al., 2020). They have been extended to dynamic scenes and human faces (Pumarola et al., 2021; Park et al., 2021a; Gafni et al., 2020; Park et al., 2021b; Chakkera et al., 2024). They usually represent a human face by sampling 3D points in a canonical space, which can be conditioned on 3DMM expression parameters to enable animation. Although they produce high-quality reconstructions, they require expensive identity-specific training. Subsequent works (Zielonka et al., 2023; Duan et al., 2023) propose techniques to reduce the training and inference time. 3DGS (Kerbl et al., 2023) became very popular as it achieves real-time rendering with high visual quality, by representing complex scenes with 3D Gaussians. It has recently been applied for dynamic human avatars (Cho et al., 2024; Qian et al., 2024; Xu et al., 2024c; Dhamo et al., 2024; Wang et al., 2025). However, most approaches learn identity-specific models. Very few recent works propose generic models (Chu and Harada, 2024; Chu et al., 2024; Kirschstein et al., 2024b), which preserve the high-quality rendering of 3DGS, while trained on a large-scale dataset of multiple identities, enabling generalization to unseen human faces. However, generic models learn a general domain prior and usually fail to produce unique identity-specific facial details, such as wrinkles or tattoos, as studied in this paper.

**Personalization.** Numerous works have proposed ways to adapt pre-trained models to various downstream tasks (Houlsby et al., 2019; Zhang et al., 2023b). Parameter-efficient fine-tuning (PEFT) techniques are proposed to fine-tune large models efficiently. LoRA (Hu et al., 2022) adds low-rank matrices into each layer of a pre-trained model, leading to a significant decrease of the learnable parameters and on-par performance compared to fine-tuning the entire model. In the context of face animation, fine-tuning part of the model has been utilized (Chatziagapi et al., 2024; Li et al., 2025), as well as meta-learning. For instance, MetaPortrait (Zhang et al., 2023a) adopts a meta-learning approach to allow adaptation during inference, while Gao et al. (2020) uses meta-learning to adapt a NeRF to a single image of an unseen subject. Moreover, MyStyle (Nitzan et al., 2022) personalizes a pre-trained StyleGAN by fine-tuning regions of its latent space, using a set of images from an individual. Similarly, One2Avatar (Yu et al., 2024) adapts a generic NeRF to one or a few images of a person. TalkLoRA (Saunders and Namboodiri, 2024) applies LoRA for the task of 3D mesh animation, while My3DGen (Qi et al., 2025) adapts LoRA to the convolutional layers of StyleGAN2 in an EG3D-based network (Chan et al., 2022). Due to its popularity and efficiency, we study LoRA for our generic model for head avatar animation. However, we find that LoRA is not sufficient to capture high-frequency facial details of a new identity. Thus, we propose to improve personalization by learning an additional register module, inspired by register tokens in ViTs.

**Additional Tokens in Neural Networks.** Memory augmentation in neural networks goes back to long short-term memory (LSTM) units (Hochreiter and Schmidhuber, 1997) that store information through gates. Memory networks (Weston et al., 2014; Sukhbaatar et al., 2015) have access to external long-term memory. More recently, transformers have emerged as a powerful representation

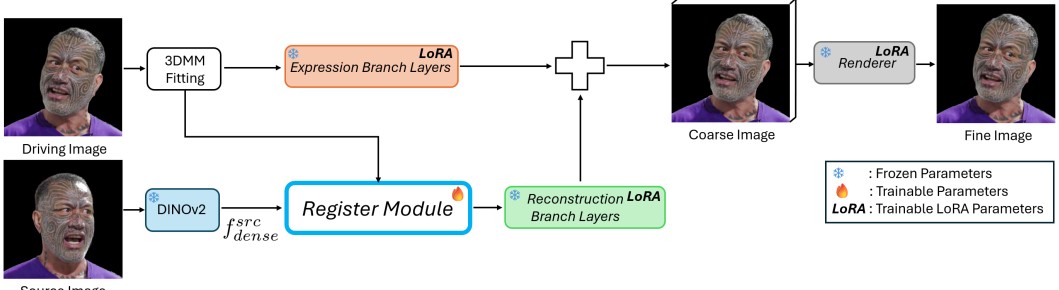

Figure 2: Illustration of our Register Module in a generic avatar animation model. During adaptation, we pass the source image's DINOv2 features $f^{src}_{dense}$ and driving image's 3DMM parameters to our module. Our module teaches the model to attend to specific regions in the dense DINOv2 features, thus providing better learning signals for LoRA to capture identity-specific details. Note that the Register Module is not needed during inference but serves as the register during LoRA training.

for various deep learning tasks, where the core element is self-attention (Vaswani et al., 2017). For language modeling, many works extend the input sequence of transformers with special tokens. Such additional tokens provide the network with new information, e.g. [SEP] in BERT (Devlin et al., 2019), or gather information for later downstream tasks, e.g. [CLS] tokens (Dosovitskiy et al., 2021), or [MASK] for generative modeling (Bao et al., 2021). Unlike these works, Darcet et al. (2023) present additional tokens as registers for storing and repurposing global information. Inspired by this, we extend registers to a 3D feature space for human faces. We learn a Register Module that stores information about distinctive high-frequency details of a human face.

# 3 Proposed Method

Figure 2 illustrates an overview of our proposed framework that adapts a general avatar model to a particular identity. Inspired by Parameter-Efficient Fine-Tuning (PEFT), specifically LoRA (Hu et al., 2022), we utilize LoRA to adapt the weights of a generalized avatar model to a particular identity. Through experiments, initially we find that adapting with LoRA does not sufficiently improve personalization (see Figure 1). Motivated by Darcet et al. (2023) that introduce registers in ViTs to store and repurpose global information, we propose a Register Module to store information about identity-specific details. Our Register Module essentially teaches the model to attend to specific regions in the dense DINOv2 features during adaptation. Importantly, it is only used during adaptation and is deactivated at inference time. With its guidance, the model learns to leverage DINOv2 features more effectively, enabling high-quality personalized head avatar generation with real-time speed from a single source image at inference.

Our proposed pipeline for personalizing a generic avatar animation model consists of two main components:

(1) We add LoRA weights to specific pre-trained layers of a generic avatar animation model (see Sec. 3.1), to keep the adaptation parameters efficient and to avoid catastrophic forgetting (see Sec. 3.2).

(2) We design a Register Module that learns a 3D feature space, facilitating the attention to specific regions of DINOv2 features, while adapting to a face from multiple views (see Sec. 3.3).

We first describe a generic avatar animation model in Sec. 3.1. Next, we describe the process of adding LoRA weights to pre-trained layers in Sec. 3.2. Finally, we describe the architecture of our Register Module in Sec. 3.3.

## 3.1 Preliminaries: Generic Avatar Generation

An avatar generation model consists of two branches: (a) reconstruction branch, and (b) expression branch. The reconstruction branch generates an animatable head avatar from the source image. The expression branch extracts the expressions and pose from the driving image which is used to animate the generated head avatar. These branches are merged and the output is rendered using a neural renderer. This process learns a model for generalized head avatar reconstruction (see Figure 2).

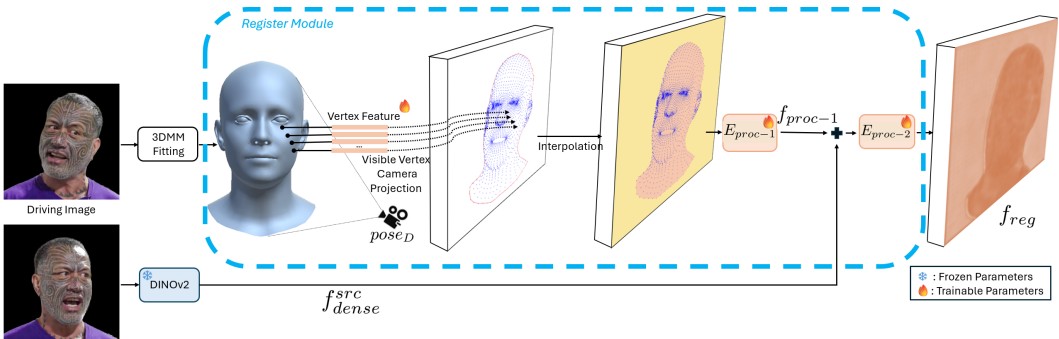

Figure 3: Illustration of our Register Module. We propose a Register Module that learns features in the 3D space. We rig embeddings to verrtices on a 3DMM mesh and use camera pose $pose_D$ to project visible vertices and their embeddings onto a camera plane. Next, we interpolate features in the face mask region, fill in the background feature. Finally, we add these features to source image's DINOv2 features $f_{dense}^{src}$ to improve learning signals for LoRA.

In particular, we use GAGAvatar (Chu and Harada, 2024) as our generic model trained on a large-scale dataset using the general DINOv2 features in its reconstruction branch, which is suitable to serve as our foundation model for fast adaptation. While DINOv2 features are robust for generic tasks, they may contain irrelevant information for avatar generation. Thus, adapting the layers of the generic avatar model to focus on relevant information within the DINOv2 feature space is necessary. We propose our Register Module for this purpose in the following sections.

## 3.2 LoRA for Fast Adaptation

To adapt to a particular identity, inspired by the literature in NLP, we use LoRA (Hu et al., 2022) for adaptation in a parameter-efficient manner. For a pre-trained weight matrix $W \in \mathbb{R}^{m \times n}$, LoRA models the adapted weights $W_{adapt}$ by representing it as an addition of the pretrained weights $W$ and an offset matrix $\Delta W$, the latter of which is low-rank decomposable.

$$W_{adapt} = W + \Delta W = W + BA, \tag{1}$$

where $B \in \mathbb{R}^{m \times r}$ and $A \in \mathbb{R}^{r \times n}$ and $r \ll min(m, n)$. During adaptation, only parameters in $A$ and $B$ receive gradients. During inference, we *merge* the offset matrix $\Delta W$ with the pre-trained weight matrix, using equation 1. Thus, there is no change in inference time. For our purpose, we add LoRA weights $A$ and $B$ to each parameter matrix in a pre-trained avatar model, except the DINOv2 model. In our implementation, for all experiments except ablations described in the supplementary material, we use the same rank $r = 32$ for all comparisons.

## 3.3 Register Module

Figure 3 illustrates the design of our Register Module. We hypothesize that, in addition to adding LoRA weights, we need a mechanism for better extraction of fine details of an identity, such as tattoos, wrinkles, muscular idiosyncrasies, and other personal features of an identity. To this end, we introduce a Register Module to improve the focus on identity-specific details.

**Feature Learning Procedure.** Specifically, we propose to highlight detailed information in DINOv2 local features of the source image with the output of the Register Module. Let $M = (V, E, F)$ be a 3DMM mesh (Li et al., 2017), where $V$ is the set of vertices, $n(V)$ is the number of vertices, $E$ is the set of edges and $F$ is the set of facets. In our Register Module, we rig embeddings $e \in \mathbb{R}^{n(V) \times D}$, where D is the dimension of the embeddings, to vertices $v \in V$ of mesh $M$. Given a driving image camera pose and position $pose_D$, we compute the set of visible vertices $U \subset V$ of the mesh $M$ from

$$U = \texttt{visible}(M, pose_D) . \tag{2}$$

Next, we project these points $U$ to a feature space in the camera plane $S = \{(i, j) \in \mathbb{Z}^2 | 1 \leq i \leq H, 1 \leq j \leq W\}$ and their corresponding embeddings to a dense feature $f_S \in \mathbb{R}^{H \times W \times D}$, where $H, W$ are the parameters of the image size, using a Perspective Projection:

$$U_S = \texttt{perspective\_project}(U, pose_D, K(S)) , \tag{3}$$

where $K(S)$ is the intrinsic camera matrix for a camera with $S$ as the camera plane. The projected points are rounded off to the nearest integer. At points in the feature plane where a visible vertex $v \in U$ is projected to, we assign the corresponding point's embedding from $e$. Hence the operation becomes:

$$f_S[u_S^i] := e[u^i] \text{ for } u^i \in U \text{ and } u_S^i \text{ is the projection of } u^i . \tag{4}$$

Given the set of points $U_s$ on the camera plane, we compute an alpha shape (Edelsbrunner et al., 1983) to find the simple contour polygon $P_{U_S}$ of the vertex projections. Let $\texttt{interior}(P)$ represent all the points inside a simple polygon $P$. For each point $p \in \texttt{interior}(P_{U_S})$ and $p \notin U_S$, we compute $k$ nearest points in $U_S$, and do inverse distance weighted interpolation for point p. Mathematically, interpolated feature $e_p$ for point $p \in \texttt{interior}(P_{U_S})$ and $p \notin U_S$ is defined as

$$f_S[p] := e_p = \frac{\sum_{i=1}^{k} \frac{1}{d_i} e_{v_i}}{\sum_{i=1}^{k} \frac{1}{d_i}} , \tag{5}$$

where $\{v_i : i \in \{1, ..., k\}, v_i \in U_S\} \subset U_S$ is the set of $k$ nearest projected vertices and $d_i = ||p - v_i||_2$. For points $p \in S$ and $p \notin \texttt{interior}(P_{U_S})$, we assign an feature $e_b$. This results in a dense constructed feature $f_S \in \mathbb{R}^{H \times W \times D}$ from assigned features at each point in $S$. We further process these features using a CNN-based encoder $E_{proc-1}$.

$$f_{proc-1} = E_{proc-1}(f_S) , \tag{6}$$

where $f_{proc-1} \in \mathbb{R}^{H \times W \times D_{out}}$.

We add these features to the source image's dense DINOv2 features $f_{dense}^{src} \in \mathbb{R}^{H \times W \times D_{out}}$ in order to exploit identity-specific information in $f_{dense}^{src}$ and process the result with another CNN-based encoder $E_{proc-2}$. Mathematically, the output of our Register Module $f_{reg}$ is

$$f_{reg} = E_{proc-2}(f_{dense}^{src} + f_{proc-1}) . \tag{7}$$

In our implementation, we use $H = W = 296$ and $D_{out} = 256$ as in Chu and Harada (2024). We set $k = 11$ and $D = 512$ for all our comparisons.

**Objective Functions.** In order to make sure that the Register Module learns meaningful features, we constrain the training with two losses. First, we use the MSE loss between the driving image's DINOv2 features ($f_{dense}^{dri}$) and output of the Register Module $f_{reg}$.

$$L_{feat} = ||f_{dense}^{dri} - f_{reg}||_2^2 . \tag{8}$$

Next, to ensure that the features learned in the Register Module are not similar to each other, we regularize the embeddings $e$. This is enforced by

$$L_{reg} = \frac{\texttt{pcos}(e)}{n(V)(n(V) - 1)} , \tag{9}$$

where $\texttt{pcos}(X) = \sum_i \sum_j \frac{X_i \cdot X_j}{||X_i||||X_j||} - n(V)$ is the sum of the non-diagonal elements of a self-pairwise cosine distance. We use a weighted combination of $L_{feat}$ and $L_{reg}$ with weights $\lambda_{feat}$ and $\lambda_{reg}$. Together, they form $L_{register} = \lambda_{feat} L_{feat} + \lambda_{reg} L_{reg}$. In our implementation, we set $\lambda_{reg} = 20$ and $\lambda_{feat} = 2$.

**Avatar Adaptation and Generation.** To adapt to a particular identity, we pick the first frame of a video as the source image and select a random frame as the driving image. Next, we predict the Register Module's output $f_{reg}$ from the source image's DINOv2 features $f_{dense}^{src}$ and driving image's 3DMM parameters. Next, we pass the driving image's 3DMM parameters to the expression branch and $f_{reg}$ to the reconstruction branch. The outputs of the corresponding branches are merged to produce a coarse image. A neural renderer then produces the fine image.

During inference, we **skip** the Register Module and directly pass the source image's DINOv2 features $f_{dense}^{src}$ to the reconstruction branch, while the rest of the process is the same as the training stage.

## 4 Experiments

### 4.1 Dataset Collection

We propose a new dataset, namely RareFace-50. Prior work uses datasets with a large number of identities, e.g. VFHQ (Xie et al., 2022). However, these datasets mostly include videos of celebrities

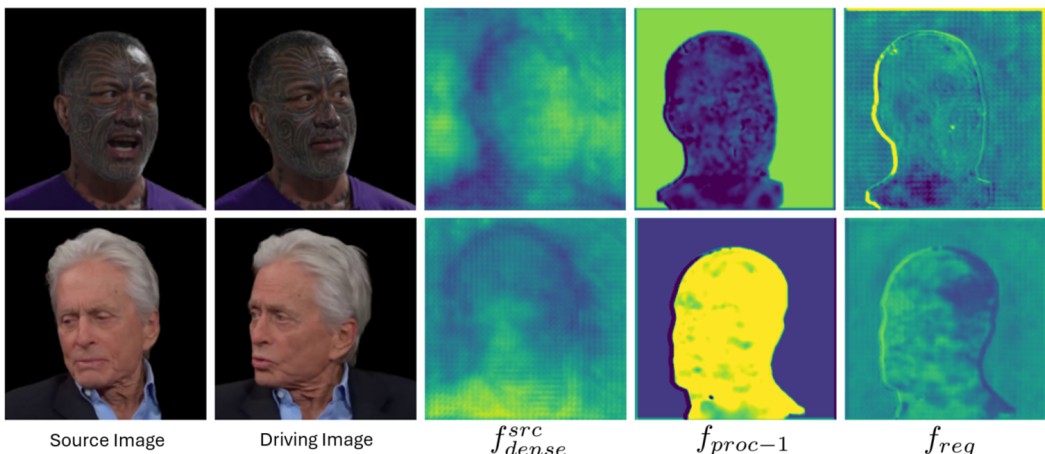

| Source Image | Driving Image | $f_{dense}^{src}$ | $f_{proc-1}$ | $f_{reg}$ |

Figure 4: Visualization of learned features by the Register Module on our RareFace-50 dataset. We visualize 1) source image's DINOv2 feature $f_{dense}^{src}$, 2) $f_{proc-1}$, output from $E_{proc-1}$, and 3) $f_{reg}$, output of the Register Module. We compute the 2nd channel-wise PCA component and standardize the values. We observe that the Register Module improves the learning signals by highlighting face regions and dampening the background regions.

and well-known faces from television. Thus, they might lack in diversity in terms of age and high-frequency facial details, such as wrinkles or unique tattoos (see Figure 4). These underrepresented human faces are difficult to faithfully generate by generic networks, such as GAGAvatar (Chu and Harada, 2024). Identifying this issue in existing datasets, we collect a video dataset of 50 identities with *unique facial details* from YouTube. The dataset is collected from high-resolution close-up videos shot in 1080p, 2K and 4K formats. We detect faces, crop and resize face images to $512 \times 512$ resolution. The average duration of the videos in this dataset is around 15 seconds, with 2 videos per identity, resulting in the total number of videos in the dataset equal to 100. We intend to publish the dataset for research purposes. In addition to RareFace-50, we use VFHQ test set and HDTF dataset to evaluate our method. HDTF (Zhang et al., 2021) consists of 362 videos each cropped and resized to 512x512 resolution. VFHQ Test consists of 50 high quality videos from 50 different identities cropped and resized to $512 \times 512$ resolution. Each video is around 4 to 10 seconds in diverse poses and settings.

We pre-process input videos using the tracking pipeline from Chu and Harada (2024). This step provides background-matted input frames, along with its tracked 3DMM parameters (these include view pose, eye pose, jaw pose, FLAME shape and expression parameters). We also pre-compute visible vertices of the 3DMM mesh fitted on a particular frame. After this, we also compute the alpha-shape polygon for the projection of the visible vertices and the set of points that lie within this polygon given the scale of the projection screen size. See more details in the supplementary material.

## 4.2 Learned Features

In Figure 4, we visualize the learned features of our Register Module. Specifically, we visualize features 1) $f_{proc-1}$ from $E_{proc-1}$, and 2) $f_{reg}$ from $E_{proc-2}$. We compute the 2nd channel-wise PCA component of DINOv2 features, and standardize and visualize using colormap between the range $[-3\sigma, 3\sigma]$ for DINOv2 features and $[-\sigma, \sigma]$ for $f_{proc-1}$ and $f_{reg}$. Since the DINOv2 model is trained in a self-supervised manner to make features of different augmented views of an input image to be similar on a diverse dataset, we observe features predicted by DINOv2 to have features *irrelevant* to the task at hand, i.e., representing human faces. Moreover, the addition of the features from $f_{proc-1}$ changes the characteristics of the added DINOv2 features (see the comparison of $f_{dense}^{src}$ and $f_{reg}$ in Figure 4), changing the distribution in irrelevant regions of the DINOv2 features. In the supplementary material, we show additional visualization results, other PCA components, and an analysis on the norms of DINOv2 and Register Module features indicating that they improve meaningfully.

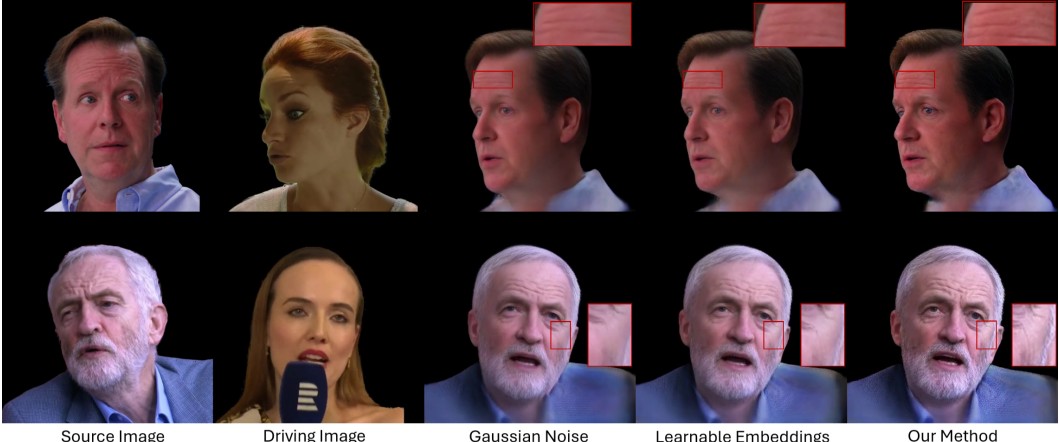

| Source Image | Driving Image | Gaussian Noise | Learnable Embeddings | Our Method |

Figure 5: Ablation results on the VFHQ Test dataset. We observe that our method performs better in preserving fine-details such as wrinkles and blemishes.

Table 1: Ablation study on our proposed Register Module on VFHQ Test dataset. In (a), we present the results with vanilla LoRA adaptation. In (b), we add noise to DINOv2 features during LoRA adaptation. In (c), we add learnable embeddings to DINOv2 features during LoRA adaptation. In (d), replace learnable embeddings $e$ rigged to vertices of the mesh with gaussian noise vectors. Best results are highlighted in **bold**.

| Method | LPIPS↓ | ACD↓ |
|---|---|---|
| (a) LoRA | 0.2666 | 0.3687 |
| (b) LoRA + Gaussian Noise | 0.2699 | 0.3795 |
| (c) LoRA + Learnable Embeddings | 0.2622 | 0.3604 |
| (d) LoRA + Gaussian Noise Vertex Embeddings | 0.2493 | 0.3623 |
| Ours (LoRA + Register Module) | **0.2470** | **0.3559** |

## 4.3 Ablation Study

We conduct an ablation study on our Register Module comparing our method with variants. This study shows the contribution of our proposed 3D feature space, which extends the idea of registers to 3D human faces. Specifically, in variant (b) LoRA + Gaussian Noise, we sample gaussian noise $G = (G_{h,w,d}) \in \mathbb{R}^{H \times W \times D_{out}}$, $Z_{h,w,d} \overset{\text{iid}}{\sim} \mathcal{N}(0,1)$ and add this noise $G$ to the features $f_{dense}^{src}$ during the LoRA adaptation stage. In variant (c) LoRA + Learnable Embeddings, we add a learnable embedding dictionary $e_{learn} \in \mathbb{R}^{H \times W \times D_{out}}$ to $f_{dense}^{src}$ and make it trainable during LoRA adaptation. In variant (d) LoRA + Gaussian Noise Vertex Embeddings, we replace the learnable embeddings $e$ rigged to the vertices of the mesh with Gaussian noise in our Register Module. In all variants, we directly use $f_{dense}^{src}$ during inference.

Figure 5 shows visual results from these variants. Adding Gaussian Noise causes washed out colors and overly smoothed details. Adding Learnable Embeddings improves the preservation of details and colors slightly. This variant would be the immediate extension of registers from Darcet et al. (2023) to our case. However, we notice that our proposed Register Module best preserves high-frequency details, such as roughness of the face and fine wrinkles, by learning an appropriate 3D feature space for human faces. Table 1 shows the corresponding quantitative results, demonstrating the efficacy of our module in enhancing identity-specific details. We see that our method has better perceptual similarity to the input image and better preserves identity. We encourage the readers to watch our supplementary video for additional results demonstrating the efficacy of our Register Module.

## 4.4 Evaluation

**Qualitative Evaluation.** Figure 6 shows our qualitative results. Notice how our method faithfully reconstructs fine details and identity-specific features well as compared to other methods. GAGAvatar

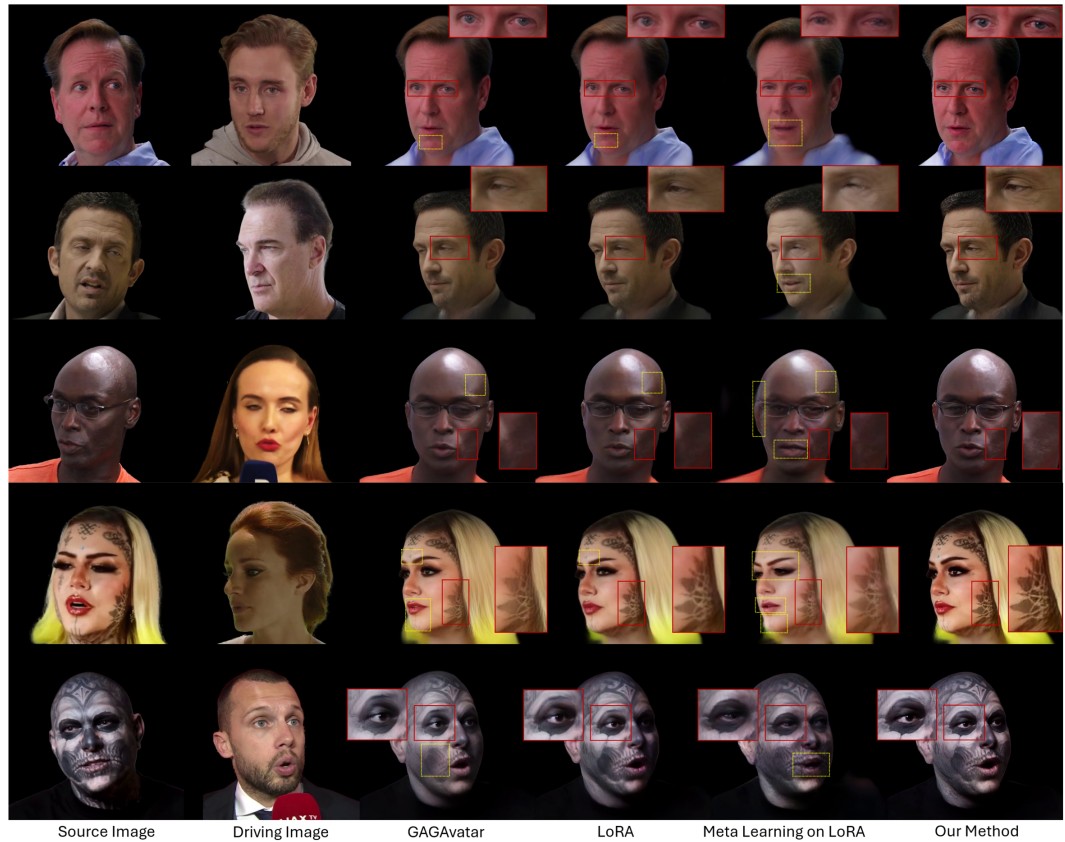

| Source Image | Driving Image | GAGAvatar | LoRA | Meta Learning on LoRA | Our Method |

Figure 6: Personalized head avatar generation on VFHQ Test (row 1, 2, and 3) and RareFace-50 (rows 4 and 5). We compare state-of-the-art methods for adaptation (LoRA (Hu et al., 2022) and Meta-Learning (Zhang et al., 2023a)). We observe that our method preserves fine details for identity-specific features and produces higher quality results as compared to other methods.

Table 2: Quantitative comparisons of our approach with the baseline and other state-of-the-art adaptation methods. Results are highlighted as follows: Best and Second Best .

|  | HDTF | | VFHQ Test | | RareFace-50 | |
|---|---|---|---|---|---|---|
| Method | LPIPS↓ | ACD↓ | LPIPS↓ | ACD↓ | LPIPS↓ | ACD↓ |
| GAGAvatar (Chu and Harada, 2024) | 0.1747 | 0.3441 | 0.2540 | 0.3631 | 0.2953 | 0.4046 |
| LoRA (Hu et al., 2022) | 0.1770 | 0.3553 | 0.2666 | 0.3687 | 0.2855 | 0.3872 |
| MetaPortrait (Zhang et al., 2023a) | 0.1901 | 0.3559 | 0.2650 | 0.4131 | 0.2901 | 0.4504 |
| Ours | 0.1618 | 0.3156 | 0.2470 | 0.3559 | 0.2744 | 0.3792 |

frequently produces washed out colors (in rows 4 and 5) and muted details (rows 1, 2, and 3). While LoRA performs better than GAGAvatar, it still misses fine wrinkles and veins (rows 1, 2, and 3), contrast in tattoos and skin (rows 4 and 5), and bumpy skin (row 3). Meta Learning on LoRA weights produces artifacts on face (row 3) and in eyes (rows 1, 2, and 5), produces wrong expression as compared to the driving image (all rows), misses tattoos on skin (row 4), and generates wrong colored lips. In general, our method learns to preserve high-frequency details in the source identity and produce higher quality results while using the same number of parameters as other methods.
**Baselines.** We compare our method to a baseline as the generic avatar generation model (Chu and Harada, 2024) and the state-of-the-art approaches, namely LoRA (Saunders and Namboodiri, 2024) and MetaPortrait (Zhang et al., 2023a) for adaptation in the cross-reconstruction setting. We use the same rank $r = 32$ for all our comparisons. Note that we implement the meta-learning algorithm from MetaPortrait (Zhang et al., 2023a) on LoRA weights for fair comparisons.

Table 3: Additional comparisons with DoRA. In (a) and (b), we present results on LoRA and LoRA + Register Module (Ours) respectively. In (c) and (d), we present results on DoRA (Liu et al. (2024)) and our Register Module with DoRA.

| Method | LPIPS↓ | ACD↓ |
|---|---|---|
| (a) LoRA | 0.2666 | 0.3687 |
| (b) LoRA + Register Module | **0.2470** | **0.3559** |
| (c) DoRA | 0.2685 | 0.3853 |
| (d) DoRA + Register Module | 0.2668 | 0.3702 |

**Evaluation Metrics.** To measure visual quality, we select challenging patches with high-frequency details from predicted frames and compare them against source image patches using Learned Perceptual Image Patch Similarity (LPIPS) (Zhang et al., 2018) in the cross-reconstruction setting. Furthermore, we estimate the identity preservation using the Average Content Distance (ACD) metric (Vougioukas et al. (2019)), by calculating the cosine distance between ArcFace (Deng et al., 2019) face recognition embeddings of synthesized and source images. Essentially, the idea is that the smaller the distance between those embeddings, the closer are the synthesized images to the input source images in terms of identity.

**Quantitative Evaluation.** Table 2 shows our quantitative results. Our method significantly outperforms the state-of-the-art in low-rank adaptation (LoRA and Meta Learning on LoRA) in terms of visual quality (LPIPS) and identity preservation (ACD). We encourage the readers to watch our supplementary video for additional results demonstrating the efficacy of our Register Module.

### 4.5 Generalizability

We test the generalizability of our proposed Register Module on other LoRA-like PEFT methods such as DoRA (Liu et al. (2024)) in Table 3. Unlike LoRA, DoRA decomposes the adapted weights $W_{adapt}$ into direction and magnitude components. Its equivalent of Eq. (1) is

$$W_{adapt} = m\frac{W + BA}{||W + BA||_c},$$ (10)

where $|| \cdot ||_c$ is the vector-wise norm of the matrix, along each column $W \in \mathbb{R}^{m \times n}$ is the pretrained weight matrix, $m \in \mathbb{R}^{1 \times n}$ is a learnable direction vector initialized as $||W||_c$, and $B \in \mathbb{R}^{m \times r}$ and $A \in \mathbb{R}^{r \times n}$ are the learnable low-rank matrices with $r \ll min(m, n)$.

We find that, while (c) DoRA performs worse than (a) LoRA in preservation of high-frequency information and identity, introducing our register module during adaptation (d) improves its performance. This shows that our Register Module is compatible with various LoRA-like methods.

## 5 Conclusion

In conclusion, we introduce a novel method for personalized head avatar generation. State-of-the-art approaches for adaptation such as vanilla LoRA and meta-learning fail to preserve high-frequency details and identity-specific features. We propose a novel Register Module that enhances the performance of LoRA, by teaching the layers to attend to specific regions in the intermediate features of a pre-trained model. To demonstrate the effectiveness of our method, we collect a dataset of talking individuals with distinctive facial features, such as wrinkles and tattoos. Our method outperforms existing methods qualitatively and quantitatively, faithfully capturing unseen identities.

**Limitations and Future Work.** Although our Register Module successfully captures distinctive facial details, it might produce suboptimal results for extreme side or back views that are rarely or not at all seen in a video. In the future, we plan to extend our work to faithfully animate avatars from such rare views of individuals.

**Acknowledgments** This research was supported in part by NSF grant IIS-2212046.

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

# Appendix

The appendix is organized as follows:

We strongly encourage the readers to watch our supplementary video.

## A  Additional Ablation Study

Table 4: Ablation study on losses to adapt with our method. In (a), we remove $L_{feat}$ during adaptation. In (b), we remove $L_{reg}$ during adaptation.

| Method | LPIPS↓ | ACD↓ |
|---|---|---|
| (a) Ours w/o $L_{feat}$ | 0.2574 | 0.3604 |
| (b) Ours w/o $L_{reg}$ | 0.2508 | 0.3574 |
| Ours | **0.2470** | **0.3559** |

Table 5: Ablation study on the effect of $\lambda_{reg}$, the weight of $L_{reg}$ to adapt with our method on the VFHQTest dataset.

| $\lambda_{reg}$ | LPIPS↓ | ACD↓ |
|---|---|---|
| 1 | 0.2520 | 0.3616 |
| 8 | 0.2502 | 0.3606 |
| 16 | 0.2485 | 0.3594 |
| 20 | **0.2470** | **0.3559** |
| 32 | 0.2500 | 0.3663 |

Table 6: Ablation study on length of videos to adapt the head avatar. We set the adaptation video length to 4, 2, and 1 seconds.

| Method | LPIPS↓ | ACD↓ |
|---|---|---|
| (a) Ours w/ 4 sec | **0.2471** | **0.3639** |
| (b) Ours w/ 2 sec | 0.2471 | 0.3656 |
| (c) Ours w/ 1 sec | 0.2477 | 0.3687 |

We conduct additional ablation studies on our proposed method. Specifically, we ablate the various losses we propose. In Table 4(a), we remove the loss $L_{feat}$ to supervise the output of our register module during adaptation, and in (b) we remove $L_{reg}$ to make the learned embeddings in our Register Module different from each other. We see that removal of these losses causes a drop in performance as compared to when both losses are present. We also ablate on the hyperparameter $\lambda_{reg}$, the weight of $L_{reg}$, in Table 5. We find that $\lambda_{reg} = 20$ gives the best results. Further, we ablate on the length of the videos used to adapt our head avatars in Table 6. We see that reducing the length of the adaptation video causes a drop in the performance of the method. Note that we trim the original videos to the first 4, 2, and 1 second for these experiments.

For completeness, we also tried the case when our Register Module is used during inference; however it harms the output quality - we see an increase in the ACD metric by around 5%.

# B Implementation Details

## B.1 Addition of LoRA to layers

We use minLoRA (Chang and Kelly) library to add LoRA (Hu et al., 2022) parameters to all layers of a pretrained pytorch model. During training LoRA is instantiated as separate parameters $B \in \mathbb{R}^{m \times r}, A \in \mathbb{R}^{r \times n}$ and $r \ll min(m, n)$ from the pretrained parameters $W_{pre} \in \mathbb{R}^{m \times n}$ of the module, so that these parameters can be trained. During inference the LoRA parameters are *merged* with the pretrained parameters and assigned as the new pretrained parameters according to:

$$W_{adapt} := W_{pre} + \Delta W = W_{pre} + BA. \tag{11}$$

This ensures that the LoRA layers add no overhead to the pipeline during the inference. Given a pretrained GAGAvatar model, we add LoRA weights to all layers except in the DINOv2 feature extractor.

## B.2 Dataset Preprocessing

Given the expression code, shape code, and camera pose predicted during 3DMM fitting, we predict a 3DMM mesh. We compute visible vertices of 3DMM mesh from the computed camera pose of a particular frame using trimesh's RayMeshIntersector implementation. Specifically, we cast rays from the camera origin to each point in the mesh and compute whether any line intersects the mesh (if an intersection exists, the point is not visible). Given these visible points in 3D space, we find a screen space camera projection on a screen of the same size as DINOv2 feature space ($H = W = 296$) using Perspective Projection. Then we find an alpha-shape of these projected points with $\alpha = 0.065$. Next, we compute all the points in the alpha-shape polygon using a parallelized point-in-polygon test. For all points in the polygon that are not projected points, we also compute the $k$ nearest projected points and distances from those projected points, where $k = 11$.

## B.3 Register Module Details

The embeddings rigged to vertices on the 3DMM mesh (Li et al., 2017) are modeled in a 3D space in our register module, However, these points are projected onto a 2D space using a camera projection following which, we interpolate the features using a weighted sum of the $k$ nearest neighbors to fill up the face region in densely constructed feature $f_S$. Using the entire set of the vertices would cause all points to be projected onto the densely constructed feature $f_S$, thus impacting the interpolation process (projection of points from the back of the head might be in the $k$ nearest neighbors of a point $p \in interior(P_{U_S})$). Thus, we only project visible points from any given view in our register module.

We model $E_{proc-1}$ as a convolutional module, with 4 convolutional layers of channel sizes $[512, 512, 256, 256]$ and kernel sizes set to 3 for each layer. $E_{proc-2}$ is also a convolutional module with 4 layers of channel sizes set to 256. The first 3 layers have a kernel size of 3, while the last layer has a kernel size of 1.

## B.4 Training Details

### B.4.1 Our Method

We initialize embeddings $e$ using Xavier Normal initialization (Glorot and Bengio, 2010). We adapt head avatars with our method for a total of 1000 iterations. The batch size is set to 2. We use Adam (Kingma and Ba, 2017) optimizer with learning rate set to $1e-4$ for the LoRA layers whereas the learning rate is set to $1e-3$ for parameters in the Register Module. We use a linear learning rate scheduler with a start factor of 1.0 and an end factor of 0.1 at the 1000th iteration. Along with our proposed losses, we also keep the losses proposed by GAGAvatar (Chu and Harada (2024)), namely, RGB losses between predicted image and driving image, a perceptual loss between the predicted images and driver image, and $L_{lifting}$, loss between predicted points from reconstruction branch and vertices of the 3DMM mesh fitted on the driving image. Our adaptation takes $\approx 35$ minutes on an RTX A5000 GPU, consuming $\approx 23$GB of VRAM. During inference, we load and merge the LoRA weights into their corresponding layer parameters. Thus, there is no overhead during inference, i.e., it consumes the same amount of resources as GAGAvatar during inference.

### B.4.2 Baselines and Ablations

For all comparisons with the vanilla LoRA, we use the same hyperparameters as our method. That is, we adapt with vanilla LoRA for a total of $1000$ iterations, set the batch size to 2, use Adam optimizer with learning rate set to $1e-4$ for the LoRA layers, and use a linear learning rate scheduler with a start factor of 1.0 and an end factor of 0.1 at the 1000th iteration. This adaptation takes $\approx 25$ minutes on an RTX A5000 GPU, consuming $\approx 14.9$GB of VRAM.

Following MetaPortrait (Zhang et al., 2023a), we implement Reptile (Nichol and Schulman (2018)), a MAML based strategy for meta-learning in our low-rank adaptation task. We perform pre-training on our RareFace-50 dataset. Following the formulation of MetaPortrait, we formulate the task of adapting to a particular identity as an inner loop task. Thus, we adapt to a randomly sampled identity at each outer step. For all comparisons with Meta-Learning on LoRA, we set rank $r = 32$, the inner loop learning rate to $2e-4$, and the outer update step size to $2e-5$. The number of inner loop steps are set to 120, and number of elements in batch in inner loop is set to 4. We set the number of outer iterations to 4800. Given resource constraints, we implement a single GPU version of Reptile (Nichol and Schulman, 2018), thus taking 12 days to complete the pretraining task on a Quadro RTX 8000 GPU, consuming $\approx 45$GB of VRAM. After the pre-training task, we adapt the model on an identity for 120 steps with the same learning rate as the inner loop, which takes $\approx 4$ minutes on an RTX A5000 GPU consuming $\approx 14.9$GB of VRAM. We then use these adapted weights for inference by merging these LoRA weights to the corresponding layers.

For all experiments with learnable parameters $e_{learn}$ as a replacement to our Register Module, we set the learning rate for $e_{learn}$ parameters to be $1e-3$.

## B.5 Metrics

We compute the visual quality metrics namely, LPIPS (Zhang et al. (2018)), on specific challenging crops with high frequency details from predicted frames and compare them against source image patches. The identity preservation metric (ACD) is measured using ArcFace (Deng et al. (2019)), a ResNet50-based network trained on WebFace (Zhu et al. (2021)). Specifically, we used "buffalo_l" model from the insightface repository (Guo et al.).

We compute the image metrics between source image and predicted image on challenging high-frequency feature crops, that include unique facial details of individuals and are the focus of our research. Since there might be a slight misalignment between the generated and ground truth head poses (because of the generation), we find that pixel-based (PSNR) or locality-based (SSIM) metrics to be unreliable. Specifically, PSNR captures blur, but doesn't capture high-frequency details (which is our main goal), as it still operates on a pixel-by-pixel basis, as in,

$$MSE = \frac{1}{mn}\sum_{i=0}^{m-1}\sum_{j=0}^{n-1}(x[i,j]-y[i,j])^2, PSNR = 10log_{10}\frac{255^2}{MSE},$$

where x and y are input images. Moreover, as indicated in the illustration below, and Figure 2 of Understanding SSIM (Nilsson and Akenine-Möller (2020)), we see that while perceptually indistinguishable on high-resolution screens, SSIM fails to recognize the perceptual similarity between the images. Note that below, we illustrate a 128x128 image using a 4x4 grid. However, the actual SSIM computation was done using the entire 128x128 images (this is only to illustrate the weaknesses of SSIM for perceptually similar images, using a toy example).

$$\text{SSIM}\left(\begin{bmatrix}128 & 128 & 128 & 128 & \cdots \\ 128 & 128 & 128 & 128 & \cdots \\ 128 & 128 & 128 & 128 & \cdots \\ 128 & 128 & 128 & 128 & \cdots \\ \vdots & \vdots & \vdots & \vdots & \ddots\end{bmatrix}, \begin{bmatrix}0 & 255 & 0 & 255 & \cdots \\ 255 & 0 & 255 & 0 & \cdots \\ 0 & 255 & 0 & 255 & \cdots \\ 255 & 0 & 255 & 0 & \cdots \\ \vdots & \vdots & \vdots & \vdots & \ddots\end{bmatrix}\right) = L \cdot C \cdot S = 0.0036$$

where $L = 1$, $C = 0.0036$, $S = 1$

$$\text{SSIM}\left(\begin{bmatrix} 0 & 255 & 0 & 255 & \cdots \\ 255 & 0 & 255 & 0 & \cdots \\ 0 & 255 & 0 & 255 & \cdots \\ 255 & 0 & 255 & 0 & \cdots \\ \vdots & \vdots & \vdots & \vdots & \ddots \end{bmatrix}, \begin{bmatrix} 255 & 0 & 255 & 0 & \cdots \\ 0 & 255 & 0 & 255 & \cdots \\ 255 & 0 & 255 & 0 & \cdots \\ 0 & 255 & 0 & 255 & \cdots \\ \vdots & \vdots & \vdots & \vdots & \ddots \end{bmatrix}\right) = L \cdot C \cdot S = -0.9964$$

where $L = 1,\ C = 1,\ S = -0.9964$

Therefore, we rely on LPIPS, as it focuses on perceptual differences between images; the primary task of our research.

## C  Data Collection

We collect data from Youtube of people knowingly appearing in interviews in public broadcasts with distinctive facial details, such as wrinkles or tattoos. These characteristics are under-represented in existing datasets. We will present the dataset as a set of links, along with trim times and crop position coordinates. Additionally, this dataset will be maintained using an automatic script that checks and removes links from the list that no longer exist in YouTube.

## D  Additional Results

### D.1  Additional Comparisons

In this section, we present results on an additional baseline. Specifically, we compare our method with single-identity models, specifically SplattingAvatar (Shao et al. (2024)).

Table 7: Additional comparisons with Single Identity models (e.g. SplattingAvatar (Shao et al. (2024))) on VFHQTest. In (a) and (b) we present results on LoRA and LoRA + Register Module (Ours) respectively. In (c) we present the results for SplattingAvatar (Shao et al. (2024)).

| Method | LPIPS↓ | ACD↓ |
|---|---|---|
| (a) LoRA | 0.2666 | 0.3687 |
| (b) LoRA + Register Module (Ours) | **0.2470** | **0.3559** |
| (c) SplattingAvatar (Shao et al. (2024)) | 0.4573 | 0.5238 |

We also present results for a suggested method (SplattingAvatar (Shao et al. (2024))) on VFHQ Test Set in Table 7. SplattingAvatar (Shao et al. (2024)) produces very poor performance quantitatively. We see a similar trend in our qualitative observations. In general, the head avatars produced by head reconstruction methods (such as SplattingAvatar (Shao et al. (2024)), FlashAvatar (Xiang et al. (2024)), or MonoGaussianAvatar (Chen et al. (2024)), etc.) overfit to the training set (identity and expressions). Inferring them using different expressions as input (not seen during training) leads to poor performance.

Here, we also create a distinction between single-identity head avatar models (such as SplattingAvatar (Shao et al. (2024)), FlashAvatar (Xiang et al. (2024)), or MonoGaussianAvatar (Chen et al. (2024)), etc.), generic head avatar models (such as GAGAvatar (Chu and Harada (2024)), GPAvatar (Chu et al. (2024)), Portrait4D (Deng et al. (2024a,b)), etc.), and generic-personalized models (our task in this paper): 1) Single-identity head avatar models require from-scratch training for each identity. Since they are limited to data from a single identity, they are unable to learn a generalizable representation and often fail in out-of-training-distribution cases. Further, limiting to a single identity also limits the ability to scale up the dataset size. 2) Generic head avatar models, on the other hand, are trained on large multi-identity diverse datasets. Thus, they can usually generalize to novel inputs (new identity and unseen expressions). These methods often trade-off generalizability with identity specificity, and might create "average" features when faced with out-of-distribution samples. 3) Thus, personalization of generic head avatar models is necessary to capture identity-specific details. Our method personalizes a generic head avatar model to produce distinctive, high-frequency features that generic head avatar models "average" out.

Further, we would like to distinguish our method from LAM (He et al. (2025)) is a one-shot head avatar model that produces an animatable Gaussian head avatar from a single image. Our goal in this paper is to personalize a generalized head-avatar model to a particular identity, such that high-frequency details are preserved. LAM focuses on animating a single image, while we personalize a single image animation model using a short clip. Moreover, our method is different from LAM's, which rigs embeddings to the vertices of a 3DMM's mesh with cross-attention layers to learn from DINOv2 features. In contrast, our Register Module operates in a learnable 3D feature space and projects back to the 2D space for facilitating LoRA adaptation, and is only used during training.

## D.2 Details of Adaptation

**Adaptation Duration.** In this section, we discuss the adaptation durations of our baselines and our method. Vanilla LoRA and our method take $\approx 25$ minutes and $\approx 35$ minutes to adapt respectively on an RTX A5000 GPU. Whereas, meta-learning on LoRA requires a much longer 12 day period on an RTX Quadro 8000 GPU for the pretraining objective, after which it requires $\approx 4$ minutes on a RTX A5000 GPU. However, during inference, all of these methods have the same inference times as GAGAvatar (Chu and Harada (2024)).

**Adaptation Parameters.** In this section, we discuss the number of parameters during adaptation and inference of our baselines and our method. During adaptation, we introduce 4.7M parameters as LoRA weights to the pretrained layers in all baselines. Our Register Module adds another 18.5M parameters. Thus, our method has 23.2M parameters during adaptation, which is $\approx 11\%$ of the total number of parameters (199M parameters) in GAGAvatar. During adaptation, we discard the trained Register Module, which lends to the same efficiency as GAGAvatar and other baselines.

## D.3 User Study

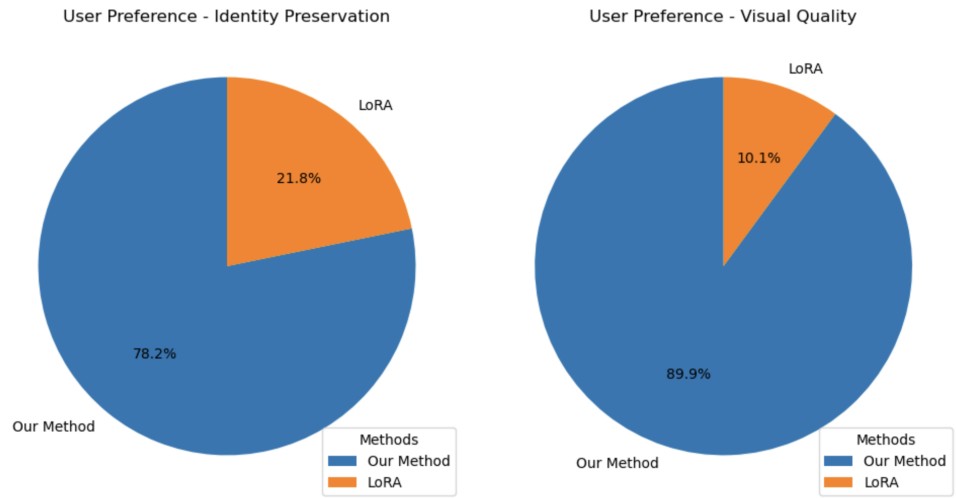

Figure 7: **User Study.** Preference (%) in terms of identity preservation, and visual quality, comparing LoRA (Hu et al., 2022), and our method.

We conduct a user study to qualitatively and subjectively compare our method against LoRA (see Sec. E for details). The results of our user study are shown in Fig. 7. We find that $78.2\%$ of the users prefer our method as compared to LoRA in terms of identity preservation. Furthermore, we find that $89.9\%$ of the users prefer our results as compared to LoRA in terms of visual quality.

## D.4 Additional Qualitative Results

In Fig. 8, we show that, compared to LoRA, our method effectively captures high-frequency details like wrinkles. We show additional results of our method against the baselines on VFHQ Test in Fig. 9 and on RareFace-50 in Fig. 10. Fig 11 and 12 show visualizations of feature norms along the channel dimensions for two identities from RareFaces-50. The values are visualized using colormaps between

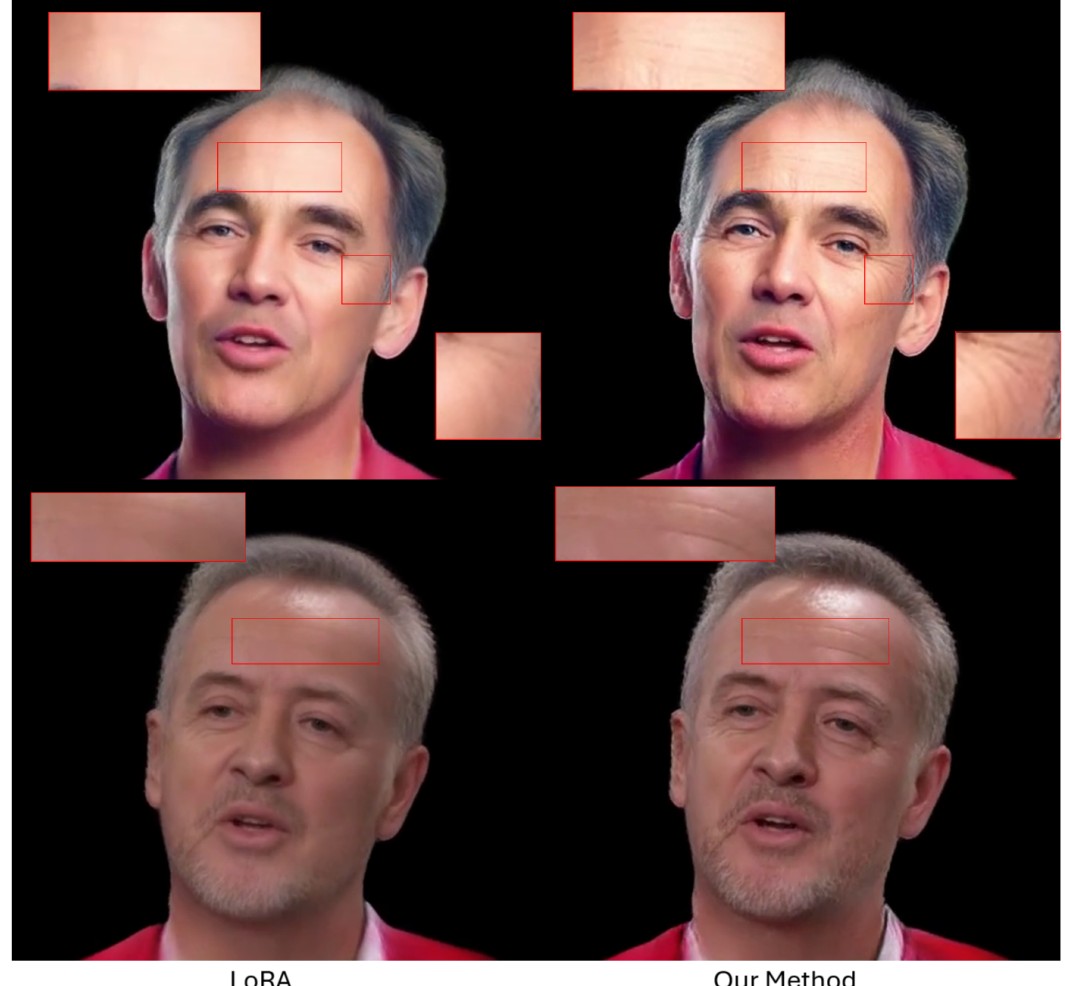

LoRA                                    Our Method

Figure 8: We show facial details that are well captured by our method with Register Module, such as wrinkles and skin folds are realistic and have higher quality than vanilla LoRA. Please note the enlarged insets of specific details.

$[-3\sigma, 3\sigma]$ for $f_{dense}^{src}$, and $f_{reg}$ and $[-\sigma, \sigma]$ for $f_{proc-1}$. We visualize the first four channel-wise PCA components in Fig. 13 to 18. We observe that the Register Module improves the learning signals by highlighting face regions and dampening the background regions.

## E    User Study Details

As mentioned in Sec. D.3, we qualitatively compare our method with vanilla LoRA as an adaptation method with a user study. We describe the details of the user study here. Fig. 19 shows the interface that we use for this user study. A total of 18 users responded to our user study. We generate head avatars given source videos from VFHQ Test and RareFace-50 using our method and LoRA. The outputs are placed side by side and the left-right orders are assigned randomly to make sure that the users are unaware of which method is ours. Each generated video is $\approx 5$ to 10 seconds long, concantenated with the source identity image and the driving video. Users are asked two questions: "Which method's avatar best looks like the source image identity?" and "Which method avatar has better visual quality?" The users can choose as answer "Method A" or "Method B" or both. Label "Method A" is placed to the left of label "Method B" and the generated videos are randomly placed in terms of a left-right order. The answers are collected through a google form. The videos are attached to the google form using a link to google drive, and the users are encouraged to download the videos

Source Image | Driving Image | GAGAvatar | LoRA | Meta Learning on LoRA | Our Method

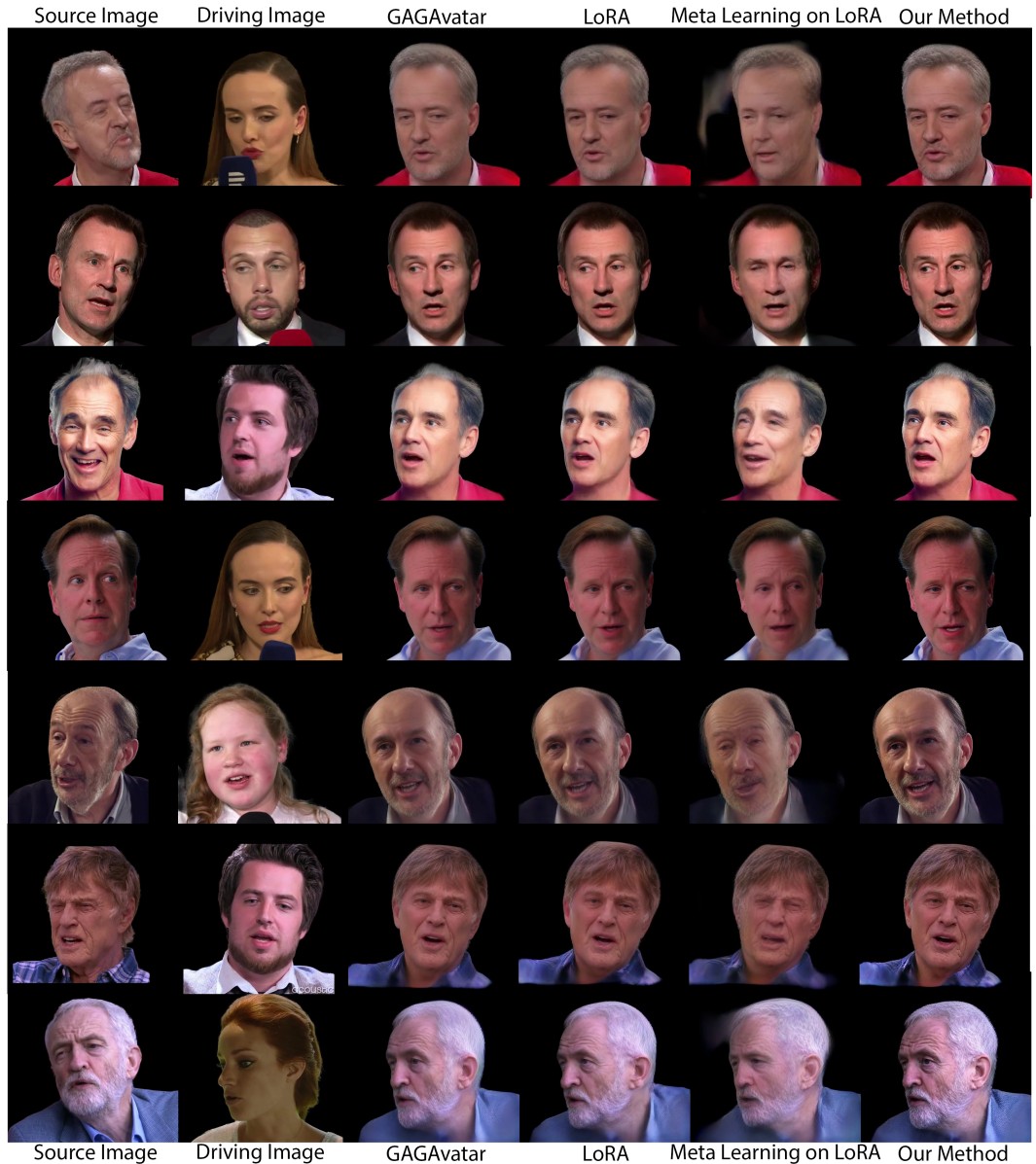

Source Image | Driving Image | GAGAvatar | LoRA | Meta Learning on LoRA | Our Method

Figure 9: Additional results of personalized head avatar generation on VFHQ Test. Please zoom in for better details.

to view them on their system. This is done to make sure that differences in high resolution are evident to the users.

## F Discussion

### F.1 Limitations

Our work is in line with lighting-agnostic methods that do not explicitly model lighting (e.g., GAGAvatar (Chu and Harada (2024)), GPAvatar (Chu et al. (2024)), Portrait4D (Deng et al. (2024a,b)), etc.) to model avatars. Therefore, lighting effects from the source image, such as highlights and shadows, are prone to being "baked" into the model's texture map or implicit appearance representation. While modeling light for avatar relighting is interesting, it is a separate line of work

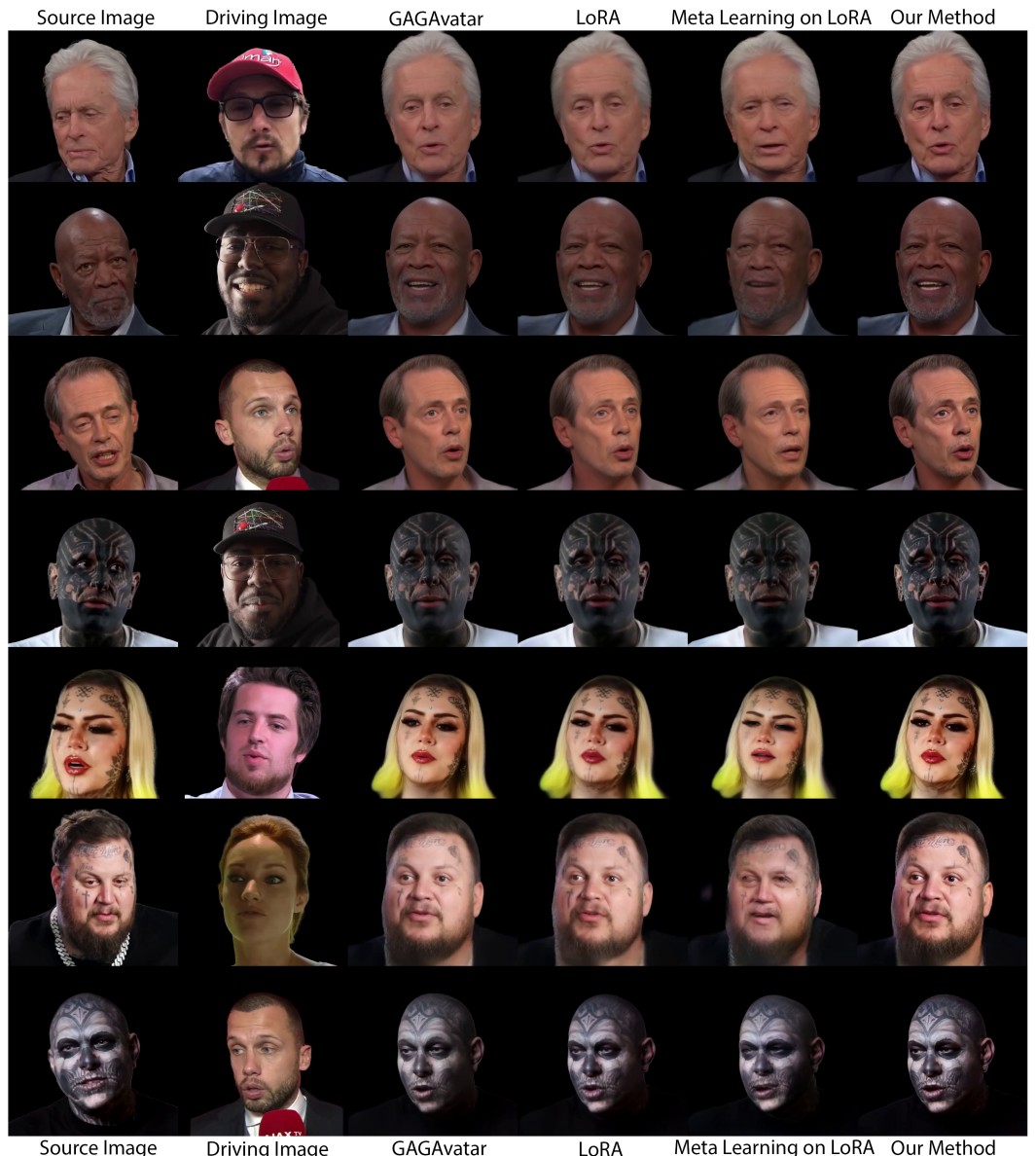

Figure 10: Additional results of personalized head avatar generation on our RareFace-50 dataset. Please zoom in for better details.

(e.g., Portrait Video Relighting (Rao et al. (2024)), Holo-Relighting (Mei et al. (2024))). We leave the task of incorporating lighting effects for the future.

Our approach does not explicitly address cloth modeling. Hence, it may not preserve clothing details. While modeling cloth characteristics and dynamics is interesting, it is a separate line of work (e.g. Guo et al. (2025)). We leave the integration of cloth modeling techniques into our approach as an interesting direction for future research.

An important factor of our method is the 3DMM fitting that is used to extract the head pose, camera parameters, and 3DMM mesh parameters (see Sec. 3.3 of the main paper). This fitting can be noisy and the error can be propagated to the final generated videos. Improving the face tracking further would be an interesting future work. Further, 3DMM fitting does not model asymmetric/extreme expressions (such as winking) and the movement of the tongue, which is another interesting line of work to pursue.

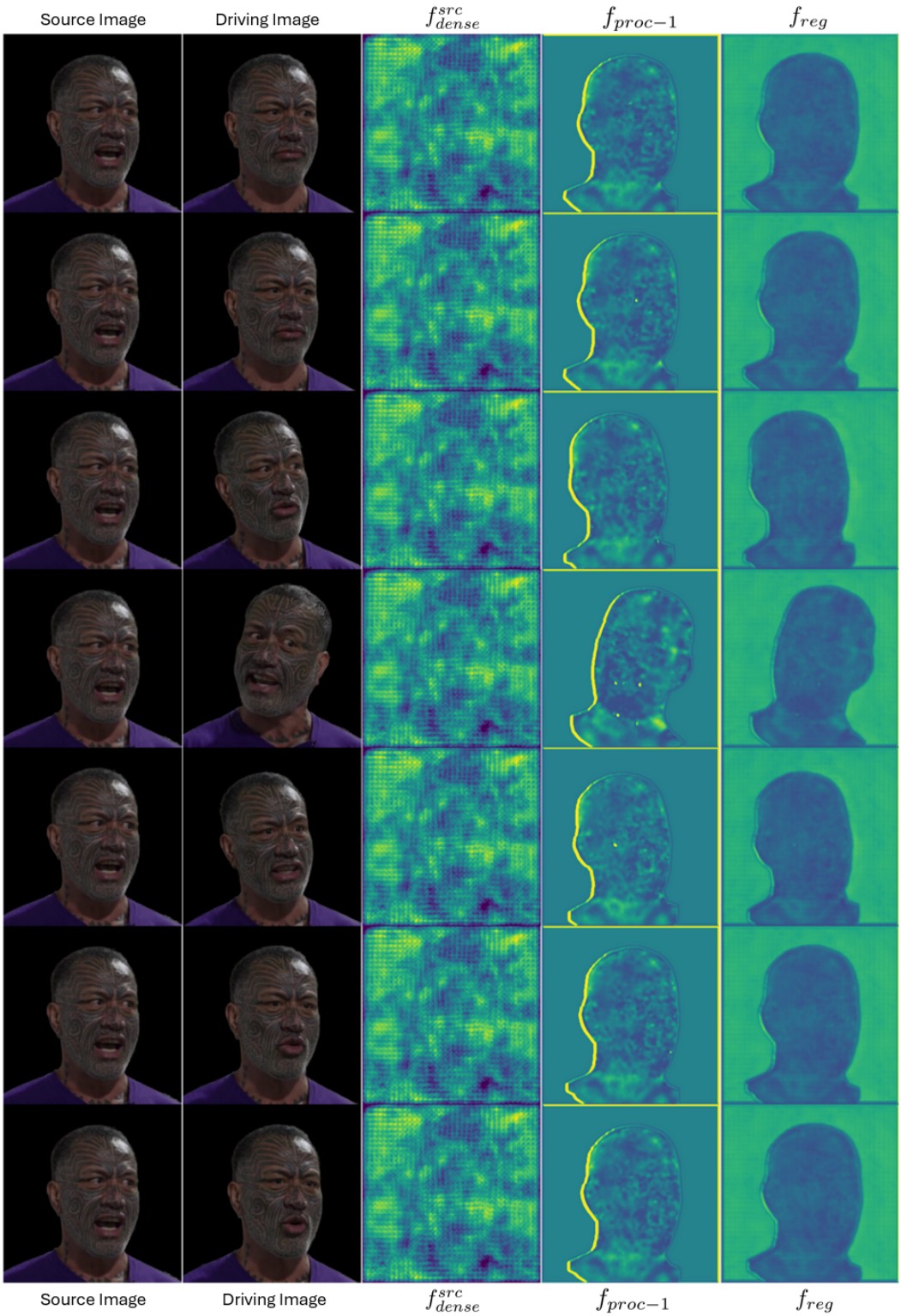

Figure 11: Visualization of learned features by the Register Module on our RareFace-50 dataset. We visualize the 1) source image's DINOv2 feature $f_{dense}^{src}$, 2) $f_{proc-1}$, output from $E_{proc-1}$, and 3) $f_{reg}$, output of the Register Module. We compute norms of the features along the embedding dimensions and standardize values.

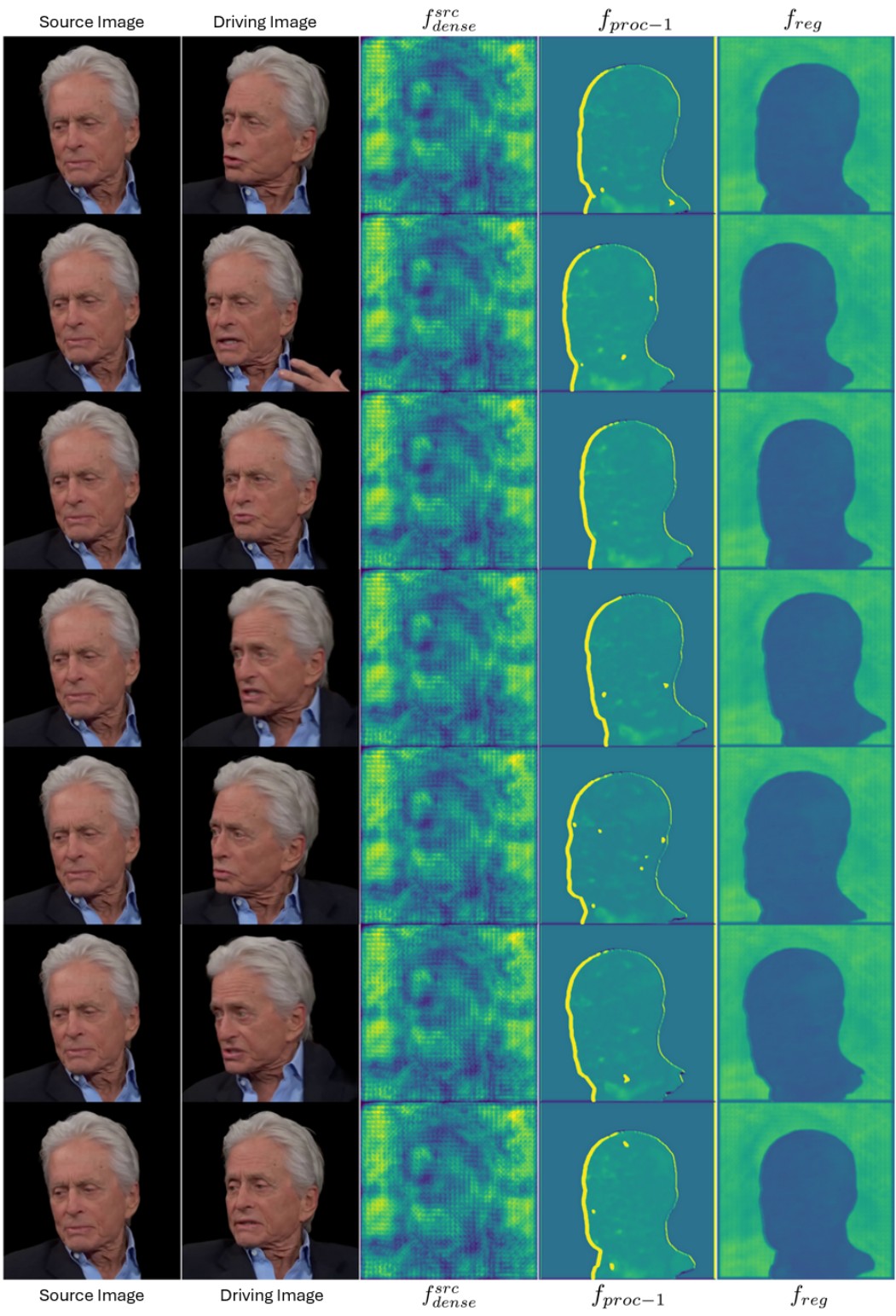

Figure 12: Visualization of learned features by the Register Module on our RareFace-50 dataset. We visualize the 1) source image's DINOv2 feature $f_{dense}^{src}$, 2) $f_{proc-1}$, output from $E_{proc-1}$, and 3) $f_{reg}$, output of the Register Module. We compute norms of the features along the embedding dimensions and standardize values.

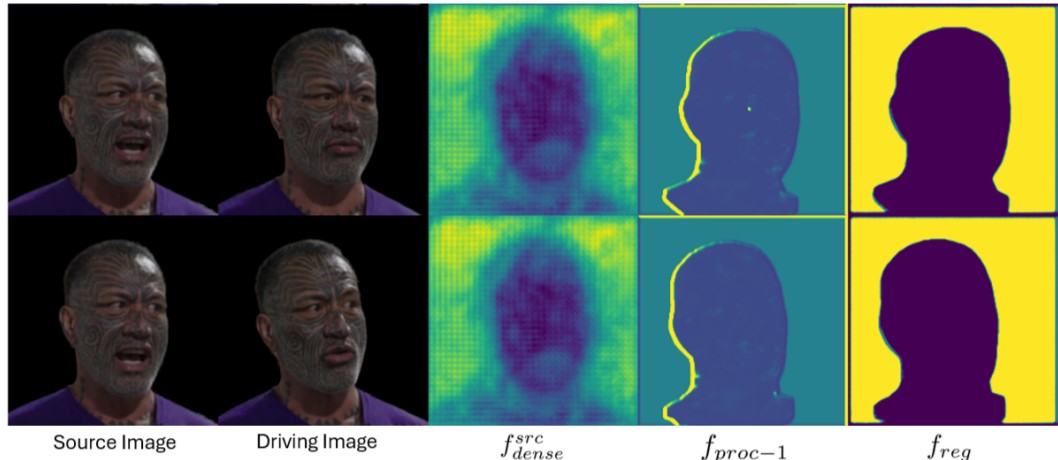

Figure 13: Visualization of learned features by the Register Module on our RareFace-50 dataset. We compute the 1st channel-wise PCA component and standardize the values.

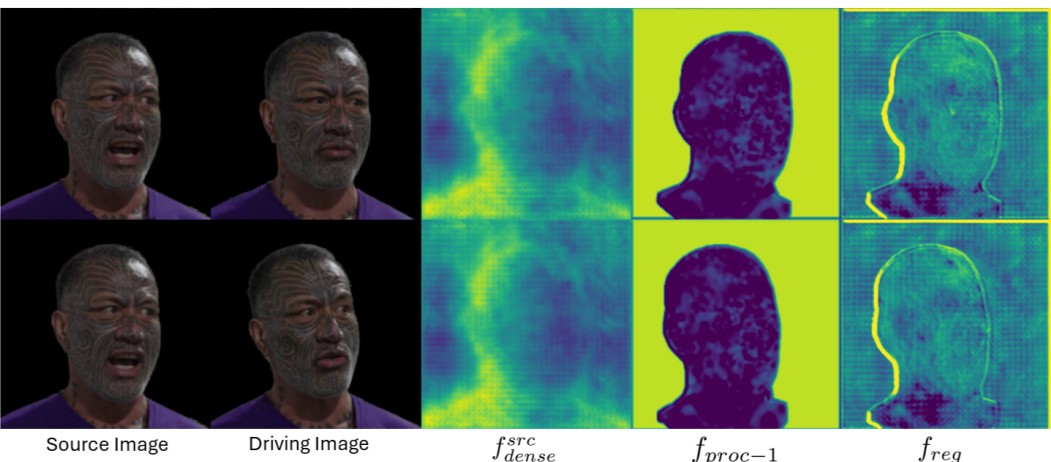

Figure 14: Visualization of learned features by the Register Module on our RareFace-50 dataset. We compute the 2nd channel-wise PCA component and standardize the values.

Our current implementation projects visible vertices and performs interpolation rather than using a rasterization-based approach. As this pipeline is not employed during inference, it does not impact the speed or scalability of the method. While we do not expect this design choice to have a significant effect on our results, incorporating a rasterization-based implementation could offer an alternative formulation, which we plan to explore in future work.

In this work, our goal is to personalize a pre-trained generic model, GAGAvatar, thus inheriting architectural choices from the original model. As these models produce videos in a frame-by-frame manner, they apply an exponential moving average (EMA) filter on expression and pose along the time axis in order to reduce temporal inconsistencies. Using their exponential smoothing, we did not observe any noticeable temporal inconsistencies in our outputs. However, temporal consistency is crucial in real-world settings and can be further explored as a future work, developing more fundamental solutions. Since 3D head avatar generation methods produce videos in a frame-by-frame manner, using tracked 3DMM parameters, there are two broad directions that can be pursued to make sure temporal consistency is fundamentally "baked-in": (a) make 3DMM parameter tracking temporally consistent (e.g., predict 3DMM parameters for a window of frames instead of one-by-one [1], etc.) and train and infer the head avatar generation model on these tracked parameters, or (b) make 3D head avatar generation temporally consistent (e.g., predict a window of future frames given

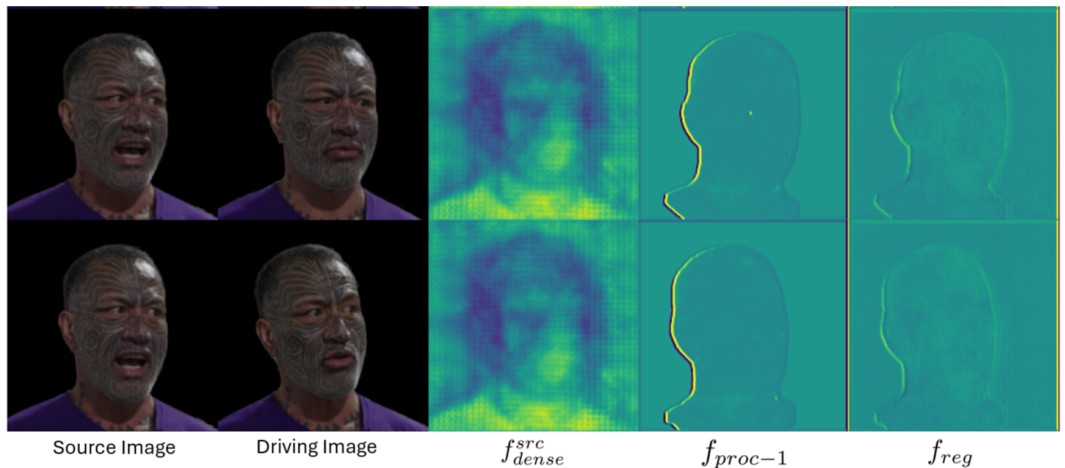

Figure 15: Visualization of learned features by the Register Module on our RareFace-50 dataset. We compute the 3rd channel-wise PCA component and standardize the values.

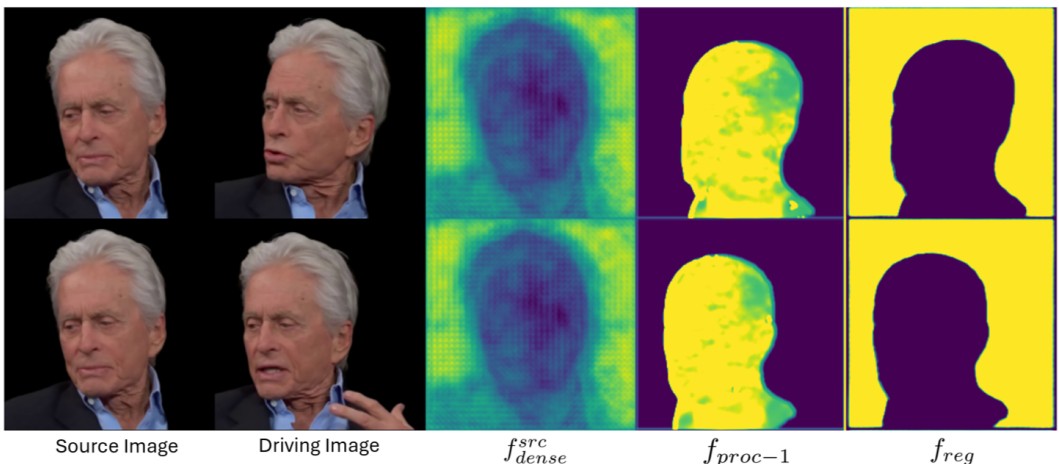

Figure 16: Visualization of learned features by the Register Module on our RareFace-50 dataset. We compute the 1st channel-wise PCA component and standardize the values.

a history of frames in a sliding window fashion [2], and/or enable temporal consistency losses using landmark tracking [3], or include other transformer-based modules etc.).

## F.2 Ethical Considerations and Broader Impacts

While our method has significant promise across diverse applications, it also carries the risk of abuse — for example, in creating "deep fakes". These can be used by users with malicious intent to spread misinformation. To prevent this, it is imperative to develop forensic tools to detect fake videos Cai et al. (2023); Reiss et al. (2023). We intend to share our code, dataset and models to improve this research, in which we will release them with strict licenses that only allow usage for academic research. When used ethically and responsibly, our method can offer profound benefits across industries — from video conferencing to the entertainment sector. In addition, we have also put appropriate procedures (see Sec. C) to ensure fair and safe use of videos from the dataset we collect.

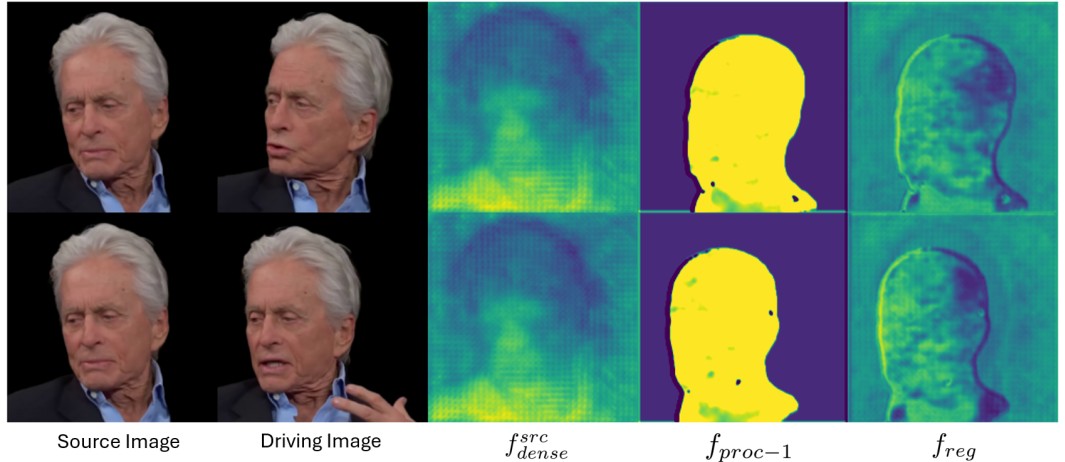

Source Image    Driving Image    $f_{dense}^{src}$    $f_{proc-1}$    $f_{reg}$

Figure 17: Visualization of learned features by the Register Module on our RareFace-50 dataset. We compute the 2nd channel-wise PCA component and standardize the values.

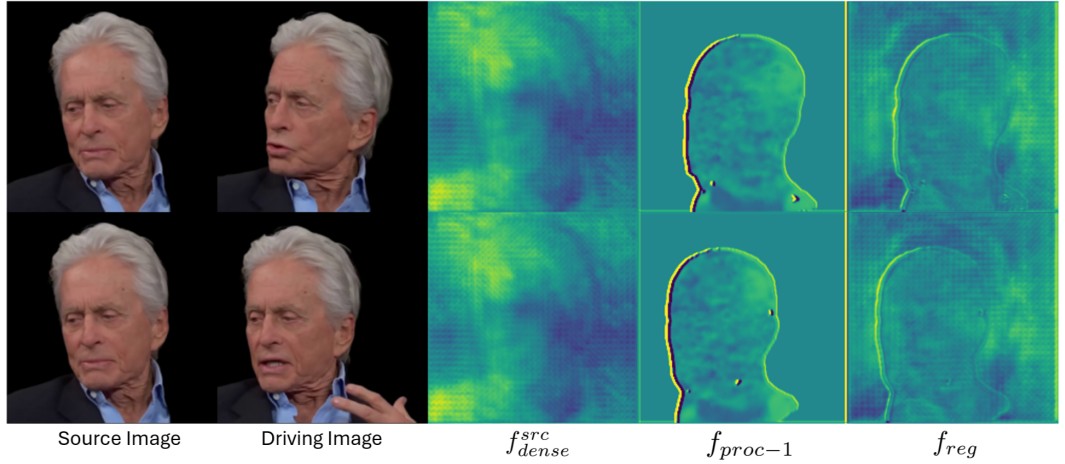

Source Image    Driving Image    $f_{dense}^{src}$    $f_{proc-1}$    $f_{reg}$

Figure 18: Visualization of learned features by the Register Module on our RareFace-50 dataset. We compute the 3rd channel-wise PCA component and standardize the values.

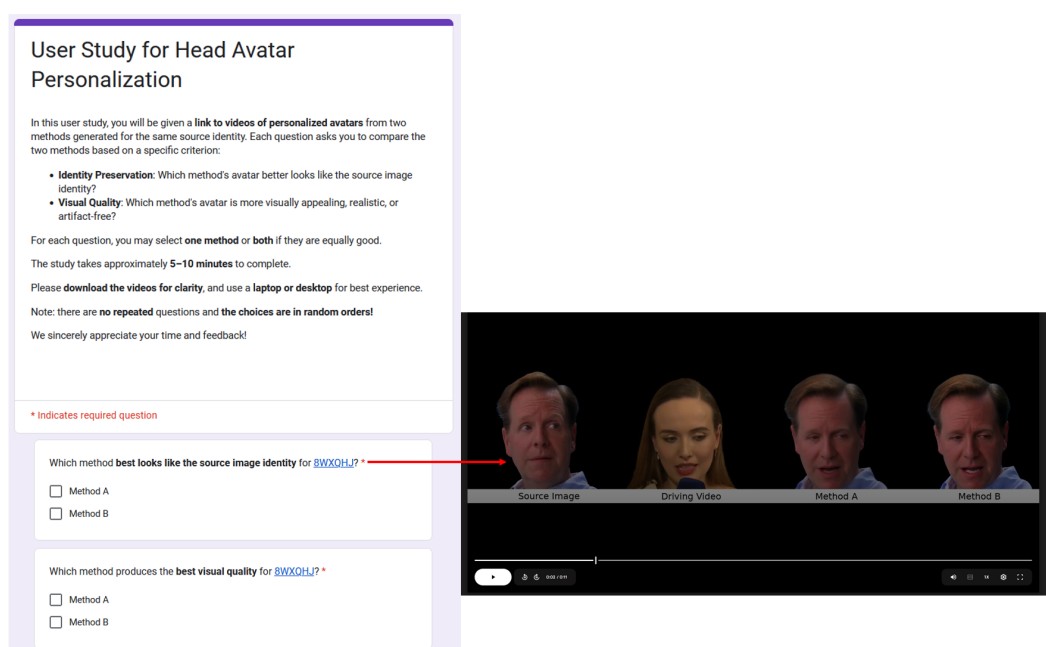

Figure 19: User Study Interface. We ask each user to watch 8 videos and answer which method preserves the source image identity and which method has the best visual quality.

