# OpenReview forum: "Low-Rank Head Avatar Personalization with Registers"
_NeurIPS.cc/2025/Conference — NeurIPS 2025 poster_

### Official Review · Reviewer_QQ5r · 2025-06-17

**Clarity:** 3
**Significance:** 2
**Originality:** 2
**Rating:** 4
**Confidence:** 4

**Summary:**

This paper tackles the problem of personalizing head avatars by adapting a pretrained generalized avatar generation model to a target subject using short videos of 4-15 seconds. The key technical contribution is a "Register Module" that enhances low-rank adaptation (LoRA) by combining local per-vertex embeddings with DINO-v2 features. The authors also contribute a video dataset of 50 subjects with distinct facial features for evaluation. They compare their method against 2 other approaches using perceptual quality (LPIPS) and identity similarity (Arcface embeddings) metrics. I would consider the potential impact as limited due to its specificity to LoRA finetuning of pretrained generalized avatar models.

**Questions:**

- Ablations
    - How does the method compare numerically to LoRA and its extensions? Specifically, what are the metrics when comparing LoRA vs. the proposed LoRA+Register Module vs. Dora vs. Dora+Register Module?
- Comparisons
    - How does the method compare against the state-of-the-art in avatar generation from videos (SplattingAvatar, FlashAvatar, and MonoGaussianAvatar)?

**Ethical Concerns:**

["NO or VERY MINOR ethics concerns only"]

**Final Justification:**

The added ablations and comparisons resolved my major concerns. I particularly appreciate the ablation of DORA and the comparison with Splattingavatar.

In summary, the paper proposes registers, which is shown to be an effective contribution to LORA and DORA and outperforms related works.

**Limitations:**

I appreciate mentioning the potential issues about imperfect 3DMM fitting (Sec. F.1). I would also include a note about the expected diversity in the input video sequence, specifically to what extent the method is robust against limited expression and head pose diversity.

**Paper Formatting Concerns:**

OK

**Quality:**

3

**Strengths And Weaknesses:**

## Strengths
+ The results with LoRA alone are indeed overly blurry and a better personalization method is needed.
+ The results after personalization improve details over the baseline (GAGAvatar)
+ The paper is easy to read.
+ The paper promises to release the collected dataset.

## Weaknesses
- Minimal Ablation
    - The ablation study is minimal and fails to analyze the most important component of the proposed method. The ablation study adds noise (a) and learnable embeddings to the DINO features f_\text{sense}^\text{src}, i.e., it studies the effect of the DINO input features (Sec. 4.3, Fig. 3). The correct component to study, however, would be the proposed register module (blue box in Fig. 3). A more meaningful ablation study could replace the learned vertex features with noise or the xyz positions or it could ablate the need for processing the rasterized features with E_\text{proc-1}. At minimum, the ablation study should provide numbers for the model variant that uses LORA vs. LORA + registers.
- Limited comparisons
    - The paper proposes a method for talking head generation from a video. There are dozens of relevant works in the literature, but the paper only compares with a feedforward method (GagAvatar) without personalization and a more generic method, which does not use a 3D Morphable Face Model (LoRA). Given the similar methodology, the comparison with GagAvatar and LoRA is important, but the comparisons should include at least the state-of-the-art for avatar generation from videos like SplattingAvatar, FlashAvatar, and MonoGaussianAvatar. Optional comparisons could be conducted by fine-tuning EG3D using Pivotal Tuning Inversion
- Missing citations
    - The paper does not cite other relevant work in avatar personalization, e.g., by leveraging meta-learning (Tancik et al. 2021), data-driven priors (Buehler et al. 2024), or extensions to LoRA (Liu et al. 2024).


## References

      Roich, Daniel, et al. "Pivotal tuning for latent-based editing of real images." __ACM Transactions on graphics (TOG)__ 42.1 (2022): 1-13.
      Chen, Yufan, et al. "Monogaussianavatar: Monocular gaussian point-based head avatar." __ACM SIGGRAPH 2024 Conference Papers__. 2024.
      Shao, Zhijing, et al. "Splattingavatar: Realistic real-time human avatars with mesh-embedded gaussian splatting." __Proceedings of the IEEE/CVF Conference on Computer Vision and Pattern Recognition__. 2024.
      Xiang, Jun, et al. "Flashavatar: High-fidelity head avatar with efficient gaussian embedding." __Proceedings of the IEEE/CVF Conference on Computer Vision and Pattern Recognition__. 2024.
      Tancik, Matthew, et al. "Learned initializations for optimizing coordinate-based neural representations." __Proceedings of the IEEE/CVF conference on computer vision and pattern recognition__. 2021.
      Buehler, Marcel C., et al. "Cafca: High-quality Novel View Synthesis of Expressive Faces from Casual Few-shot Captures." __SIGGRAPH Asia 2024 Conference Papers__. 2024.
      Liu, Shih-Yang, et al. "Dora: Weight-decomposed low-rank adaptation." __Forty-first International Conference on Machine Learning__. 2024.

---

> ### Author Rebuttal · Authors · 2025-07-31
>
> We thank the reviewer for their thoughtful feedback. We are encouraged that they found our Register Module to be a key technical contribution. We are glad the reviewer recognized the clarity of the paper, the perceptual and identity improvements over the baseline (GAGAvatar), and the value of releasing our collected dataset. We also appreciate their observation that LoRA alone produces blurry results, highlighting the importance of improved personalization strategies. We address the reviewer comments below and will incorporate all feedback in the revised manuscript.
>
> **Weakness 1 & Question 1: Ablations.**
> We thank the reviewer for their comments. We will incorporate the LoRA baseline results in our ablations table (Table 1) to highlight the difference in the ablations study. We present those results in the table below. We would like to mention that these results are present in Table 2 of the main paper in row 2 (LoRA variant) and row 4 (LoRA w/ register variant) on the VFHQ Test and RareFace-50 datasets. We also present those results in the table below.
>
> The ablation to use xyz coordinates instead of learnable embeddings for vertices in the register module is an interesting approach. Prior approaches, such as NeRF [1] and mip-NeRF [2], use multi-layer perceptrons (MLPs) and positional encoding (PE) to inflate the dimensions of the 3-dimensional input. Using an MLP+PE approach will give us a learnable embedding for each vertex, thus making it similar to our proposed approach.
>
> We present the results for the suggested ablation of replacing learnable embeddings with Gaussian noise below. We observe that using Gaussian noise embeddings leads to an improvement over variant (a) LoRA. However, they still perform worse than variant (b) LoRA w/ Reg (Ours). Qualitatively, we see higher detail preservation as compared to variant (a), but worse detail retention than our method.
>
> We also present the results with DoRA: \(c) DoRA [3] and (d) our register module with DoRA. We note that the performance of \(c) DoRA is inferior to (a) LoRA. However, introducing our register module during training improves its performance. We also see a similar trend qualitatively.
>
> Note that the constructed dense feature $f_{S}$ has $D=512$ channel dimensions (refer to Ln 288 onwards). Since $f_{dense}^{src}$ features have $256$ channel dimensions, we use $E_{proc-1}$ to scale down the dimensions of $f_{S}$ using $f_{proc-1} = E_{proc-1}(f_{S})$ to support the operation $f_{proc-1}+f_{dense}^{src}$. Hence, using $E_{proc-1}$ is necessary.
> Note: Best and second-best results highlighted in **bold** and *italics* respectively.
>
> | Method                   | LPIPS↓   | ACD↓   |
> |--------------------------|------------|------------|
> | (a) LoRA                     | 0.2666     | 0.3687     |
> | (b) LoRA w/ Reg (Ours)       | **0.2470** | **0.3559** |
> | \(c) DoRA                     | 0.2685    | 0.3853    |
> | (d) DoRA w/ Reg              | 0.2668    | 0.3702    |
> | (e) Gaussian Noise Vertex Embeddings | 0.2493    | 0.3623   |
>
>
> **Weakness 2 & Question 2: Missing Comparisons.**
> We thank the reviewers for their suggestions. We note that the papers referred to by the reviewer, SplattingAvatar [4], FlashAvatar [5], and MonoGaussianAvatar [6], are head reconstruction methods, which have a different setting from our method. Head reconstruction methods either take multi-view or single-view videos to train a NeRF or variant (e.g., Tri-Planes, Gaussian Splatting, etc), in order to learn the 3D representation of a *single identity*. In other words, they require separate from-scratch training for each identity on long videos (5-10 mins) and are sensitive to camera calibration errors. We, on the other hand, take a *generative* 3D-aware one-shot avatar model that is trained on a large-scale in-the-wild dataset and can generate any identity. We propose to personalize it using a short in-the-wild video (5-10 seconds) to preserve identity-specific and high-frequency details.
>
> **Weakness 3: Missing Citations.**
> We thank the reviewer for the relevant citations and will gladly incorporate them into our revised manuscript. Cafca [7] uses data-driven priors for novel-view synthesis of expressive faces; however, it does not generate face animations. Tancik et al. [8] use meta-learning to initialize weights in coordinate-based neural networks, with interesting applications, including image regression (but not face animation as in our paper). DoRA [3] is an extension to LoRA, which we have incorporated into our method and compared with.
>
> ## References
> [1] Mildenhall et al., ‘NeRF: Representing Scenes as Neural Radiance Fields for View Synthesis’, in ECCV, 2020.
>
> [2] Barron et al., ‘Mip-NeRF: A Multiscale Representation for Anti-Aliasing Neural Radiance Fields’, ICCV, 2021.
>
> [3] Liu et al., ‘DoRA: Weight-Decomposed Low-Rank Adaptation’, in ICML, 2024.
>
> [4] Shao et al., ‘SplattingAvatar: Realistic Real-Time Human Avatars with Mesh-Embedded Gaussian Splatting’, in CVPR, 2024.
>
> [5] Xiang et al., ‘FlashAvatar: High-fidelity Head Avatar with Efficient Gaussian Embedding’, in CVPR, 2024.
>
> [6] Chen et al., ‘Monogaussianavatar: Monocular gaussian point-based head avatar’, in ACM SIGGRAPH 2024 Conference Papers, 2024, pp. 1–9.
>
> [7] M. C. Buehler et al., ‘Cafca: High-quality Novel View Synthesis of Expressive Faces from Casual Few-shot Captures’, in ACM SIGGRAPH Asia 2024 Conference Paper, 2024.
> [8] M. Tancik et al., ‘Learned Initializations for Optimizing Coordinate-Based Neural Representations’, in CVPR, 2021.

---

> > ### Comment · Reviewer_QQ5r · 2025-08-05
> >
> > I appreciate that the authors provided method details and results on DORA. It's good to see that the new results are improved when using the proposed registers.
> >
> > It is true that the avatar reconstruction methods [4,5,6] need to optimize a 3D representation of a single identity from scratch and that they typically use minute-long videos as input. However, if I understand correctly, the proposed approach also requires an optimization for each identity on a 5-10 seconds video, which takes 35 minutes (l. 62 in the supp. PDF). I would suggest to show experimentally that the related works [4,5,6] cannot handle such short videos.

---

> > > ### Author Response · Authors · 2025-08-07
> > >
> > > Thank you for your suggestion. We present results for a suggested method (SplattingAvatar [1]) on VFHQ Test Set below.
> > > | Method          |   LPIPS↓ |    ACD↓ |
> > > |-----------------|---------|------------|
> > > | GAGAvatar       | 0.2540  | 0.3631 |
> > > | LoRA            | 0.2666  | 0.3687 |
> > > | Meta on LoRA    | 0.2650 | 0.4131 |
> > > | **SplattingAvatar** | 0.4573  | 0.5238 |
> > > | Ours        | 0.2470 | 0.3559 |
> > >
> > > As shown in the table, SplattingAvatar produces very poor performance quantitatively. We see a similar trend in our qualitative observations. In general, the head avatars produced by head reconstruction methods (such as SplattingAvatar [1], FlashAvatar [2], or MonoHeadAvatar [3], etc.) overfit to the training set (identity and expressions). Infering them using different expressions as input (not seen during training) leads to poor performance.
> > >
> > > We would like to create a distinction between *single-identity* head avatar models (such as SplattingAvatar [1], FlashAvatar [2], or MonoHeadAvatar [3], etc.), *generic* head avatar models (such as GAGAvatar [4], GPAvatar [5], Portrait4D [6], etc.), and *generic-personalized* models (our task in this paper): 1) **Single-identity** head avatar models require from-scratch training for each identity. Since they are limited to data from a single identity, they are unable to learn a generalizable representation and often fail in out-of-training-distribution cases. Further, limiting to a single identity also limits the ability to scale up the dataset size. 2) **Generic** head avatar models, on the other hand, are trained on large multi-identity diverse datasets. Thus, they can usually generalize to novel inputs (new identity and unseen expressions). These methods often trade-off generalizability with identity specificity, and might create "average" features when faced with out-of-distribution samples. 3) Thus, **personalization** of generic head avatar models is necessary to capture identity-specific details. Our method personalizes a generic head avatar model to produce distinctive, high-frequency features that generic head avatar models "average" out.
> > >
> > > ## References
> > > [1] Shao et al., ‘SplattingAvatar: Realistic Real-Time Human Avatars with Mesh-Embedded Gaussian Splatting’, in CVPR, 2024.
> > >
> > > [2] Xiang et al., ‘FlashAvatar: High-fidelity Head Avatar with Efficient Gaussian Embedding’, in CVPR, 2024.
> > >
> > > [3] Chen et al., ‘Monogaussianavatar: Monocular gaussian point-based head avatar’, in ACM SIGGRAPH 2024 Conference Papers, 2024, pp. 1–9.
> > >
> > > [4] Chu et al., Generalizable and animatable gaussian head avatar. In NeurIPS 2024.
> > >
> > > [5] Chu et al., ‘GPAvatar: Generalizable and Precise Head Avatar from Image(s)’, In ICLR 2024.
> > >
> > > [6] Deng,et al., ‘Learning One-Shot 4D Head Avatar Synthesis using Synthetic Data’, In CVPR 2024.

---

### Official Review · Reviewer_k9TX · 2025-06-22

**Clarity:** 3
**Significance:** 2
**Originality:** 3
**Rating:** 4
**Confidence:** 4

**Summary:**

The paper presents a new method for personalizing head avatar models through a LoRA approach enhanced by a learnable Register Module. This module introduces a 3D feature space that captures high-frequency identity-specific facial details such as wrinkles and tattoos, which are often missing in generic models trained on large datasets. Additionally, the authors introduce a new dataset, RareFace-50, to demonstrate the effectiveness of their approach. Quantitative and qualitative evaluations on RareFace-50 and VFHQ Test datasets show improvements over baseline and naive LoRA methods.

**Questions:**

I would like the author to answer the following questions in order to further enhance the score:

-  In the feature learning procedure starting from Line 169, why do you adopt the process of computing visible joints → projection → interpolation, instead of using a rasterization-based approach? Rasterization is a standardized and GPU-efficient method that is typically much faster and more scalable.
- Why didn’t you evaluate your method on more datasets and with additional baseline methods (other than GAGAvatar)? Is the proposed approach only applicable to GAGAvatar?

**Ethical Concerns:**

["NO or VERY MINOR ethics concerns only"]

**Final Justification:**

The rebuttal can resolve most of my concerns. However, I am still concerned about the technical contributions of this paper. Considering these, I would like to keep my rating as borderline accept.

**Limitations:**

- The proposed method still requires annotated 3DMM parameters for adaptation, which might limit its accessibility or deployment in the wild. This also makes the method inefficient for personalizing the head avatar.

- Currently, the proposed method is only demonstrated on GAGAvatar. The authors provide no evidence of its applicability to other human head avatar methods, which limits the generalizability of the approach.

**Paper Formatting Concerns:**

I have not noticed any major formatting issues.

**Quality:**

3

**Strengths And Weaknesses:**

Strengths:
- The paper is well-written and structured, making it easy to understand.
- The proposed 3D register extension is effective for the personalization training of the face avatar.
- Introduction of RareFace-50 fills a valuable gap for testing on underrepresented facial features.

Weaknesses:
- The technical novelty is limited: the approach primarily reuses LoRA, with the main contribution being the use of registers. However, the register mechanism closely resembles that of the recent SIGGRAPH paper LAM, which also leverages features from 3DMM geometry.
LAM: https://github.com/aigc3d/LAM

- In the experiments, the authors only apply the proposed method to the GAGAvatar method, while not proving the generalized applicability of the proposed method based on other head avatar methods.

- In the experiments, the authors evaluate their method on only two datasets—VFHQ and RareFace-50—without including comparisons on other relevant datasets such as HDTF (used in GAGAvatar) or Nersemble, which is now widely adopted.

HDTF: https://github.com/MRzzm/HDTF
Nersemble: https://tobias-kirschstein.github.io/nersemble/

---

> ### Author Rebuttal · Authors · 2025-07-31
>
> We thank the reviewer for their detailed and encouraging feedback. We are pleased they found our paper well-written and easy to follow, and appreciated the structure and clarity of our presentation. We are glad the effectiveness of our 3D Register Module for enhancing identity-specific facial details was recognized, as well as the meaningful role it plays in improving personalization. We also appreciate the acknowledgment of RareFace-50 as a valuable contribution that helps address the underrepresentation of distinctive facial features in current benchmarks. We address the reviewer comments below and will incorporate all feedback in the revised manuscript.
>
> **Weakness 1: Limited Novelty Due to Similarity with LAM and Reuse of LoRA.**
> LAM [1] is a one-shot head avatar model that produces an animatable Gaussian head avatar from a single image. Our goal in this paper is to personalize a generalized head-avatar model to a particular identity, such that high-frequency details are preserved. LAM focuses on animating a single image, while we personalize a single image animation model using a short clip.
>
> Furthermore, our model design is entirely different from LAM's, which rigs embeddings to the vertices of a 3DMM's mesh with cross-attention layers to learn from DINOv2 features. In contrast, our register module operates in a learnable 3D feature space and projects back to the 2D space for facilitating LoRA adaptation, and is only used during training.
>
> **Weakness 2, Weakness 3 & Question 2: Limited dataset and baseline evaluation.**
> We believe that our approach of low-rank personalization is extendable to other approaches since we use DINOv2 features, which are general domain features and commonly used in various works. We specifically chose GAGAvatar to demonstrate our method because it is a very competitive method, having been trained on a very large dataset (VFHQ [1]), and producing state-of-the-art results in the general avatar domain.
>
> VFHQ is a large-scale and high-quality video face dataset. It includes diverse identities, facial expressions, head poses, and settings, and thus it is used in recent works. On the other hand, we collected RareFace-50 to include unique facial details that are underrepresented or do not even appear in existing datasets.
> We also provide the evaluation metrics on the HDTF dataset for all baselines and our method below.
> Note: Best and second-best results highlighted in **bold** and *italics* respectively.
> | Method       |   LPIPS↓   |   ACD↓      |
> |--------------|------------|------------|
> | GAGAvatar    | *0.1747*  | *0.3441*   |
> | LoRA         | 0.1770    | 0.3553     |
> | MetaPortrait | 0.1901    | 0.3559     |
> | Ours         | **0.1618** | **0.3156** |
>
> Note that the NeRSemble [2] is a multi-view high-quality lab-setting video dataset used for the 3D multi-view face reconstruction task. This dataset is used when the method needs multi-view images and videos. We used other datasets that have in-the-wild monocular videos, which are suitable for our setting.
>
>
> **Question 1: Rasterization instead of visible vertices → projection → interpolation.**
> We appreciate this interesting suggestion as one implementation option. Since this pipeline is not used during inference, it does not affect the speed or the scalability of the method. While we believe it does not have a critical effect on our results, we will explore this option as future work.
>
> ## References
> [1] He et al., ‘LAM: Large Avatar Model for One-shot Animatable Gaussian Head’, in SIGGRAPH, 2025.
>
> [2] Kirschstein et al., ‘NeRSemble: Multi-View Radiance Field Reconstruction of Human Heads’, ACM Trans. Graph., vol. 42, no. 4, Jul. 2023.

---

> > ### Comment · Reviewer_k9TX · 2025-08-05
> >
> > Thank you authors for your detailed feedback. The rebuttal can address all of my concerns.

---

> > > ### Author Response · Authors · 2025-08-06
> > >
> > > Thank you very much for your thoughtful review and for acknowledging our rebuttal—we truly appreciate your time and constructive feedback. We are happy to address further questions if there are any.

---

### Official Review · Reviewer_wF51 · 2025-07-02

**Clarity:** 3
**Significance:** 2
**Originality:** 2
**Rating:** 4
**Confidence:** 4

**Summary:**

This paper addresses the challenge of personalizing generic, 3D-aware head avatar models to reconstruct the unique, high-frequency details of an unseen individual (e.g., wrinkles, tattoos, skin texture). The authors posit that standard adaptation techniques like Low-Rank Adaptation (LoRA) are insufficient for this task, as they tend to lose these identity-specific features.

The core contribution is a novel Register Module, designed to augment LoRA-based personalization. Inspired by registers in Vision Transformers that store global information, this module introduces a learnable 3D feature space that operates on intermediate features of a pre-trained model. The purpose of this module is to explicitly capture, store, and re-purpose the distinctive facial details of a target identity during the adaptation process.

To validate their method, the authors show through quantitative and qualitative examples that their approach outperforms standard LoRA and a meta-learning baseline in terms of both visual quality and identity preservation.

**Questions:**

**Clarification of the Register Module**: Could you please provide a more detailed architectural description of the Register Module? The CNN-based encoder is mentioned but not illustrated. A small diagram illustrating how it takes intermediate features as input, processes them through the 3D learnable space, and outputs the enhanced features would be immensely helpful.

**Quantify Efficiency Claims**: You claim the method is efficient in terms of parameters and speed. Could you please update the paper to include:

(a) The number of trainable parameters for the vanilla LoRA baseline versus your method (LoRA + Registers).

(b) The impact on inference speed (e.g., frames per second or milliseconds per frame) for both methods.


**Temporal Consistency**: It seems that the register module rely heavily on the 3DMM prediction, does the video-level prediciton is done frame-wise? How does the authors improve temporal consistency?

**Ethical Concerns:**

["NO or VERY MINOR ethics concerns only"]

**Final Justification:**

Based on the authors' thoughtful response, most of my initial concerns have been resolved, so I'm comfortable raising my score

**Limitations:**

Yes

**Quality:**

2

**Strengths And Weaknesses:**

**Strengths**:

**Well-Motivated Problem**: The paper targets a clear and significant problem. As generic avatar models become more common, the need for high-fidelity, efficient personalization is crucial. The observation that vanilla LoRA fails to capture high-frequency details is an excellent and practical motivation for this work.

**Simple and Intuitive Technical Contribution**: The proposed "Register Module" is a clever idea. Adapting the concept of "registers" from 2D Transformers to a 3D feature space for identity preservation is an insightful contribution. The intuition that LoRA's low-rank updates might be too coarse to handle fine details, thus requiring a dedicated "memory" or "cache" for them, is compelling.


**Weaknesses**:

**Lack of Architectural Detail**: The paper is surprisingly light on the specific architectural details of the Register Module. While the high-level concept is explained, it is unclear how this 3D feature space is implemented, how it interacts with the intermediate features of the DINOv2 backbone, and how information is "stored" and "re-purposed". This lack of clarity makes it difficult to fully assess the technical novelty and replicate the work from the paper alone.

**Unsubstantiated Efficiency Claims**: The abstract and introduction claim the method requires "a small number of parameters" and retains "the original inference speed." These are important claims for a personalization method but are not substantiated with data. The paper needs to report the parameter count of the Register Module and provide concrete numbers on the impact on inference time (e.g., FPS) compared to the vanilla LoRA baseline.

**Limited Discussion on Generalizability**: The method is presented and validated on a single backbone architecture (GAGAvatar). While effective, this raises questions about the generalizability of the Register Module. A discussion on whether this module is a general-purpose "plug-in" for other 3D-aware generative models would significantly broaden the paper's impact.

---

> ### Author Rebuttal · Authors · 2025-07-31
>
> We sincerely thank the reviewer for their thoughtful and detailed feedback. We are encouraged that they found our problem motivation strong and practical, particularly the need for high-fidelity personalization beyond what standard LoRA provides. We are pleased the reviewer appreciated the simplicity and intuitiveness of the Register Module, and found the adaptation of Transformer-style registers to a 3D feature space to be a novel and insightful idea. We are also glad they recognized the effectiveness of our method in capturing identity-specific high-frequency details and outperforming both standard LoRA and a meta-learning baseline. We address the reviewer comments below and will incorporate all feedback in the revised manuscript.
>
> **Weakness 1 & Question 1: Lack of Architectural Detail.**
> We thank the reviewer for their suggestion. We will highlight the architectural details and the "forward" process in the revised manuscript. We would like to note that we explain the entire "forward" process in Section 3 of the main paper and give additional architectural details of the register module in Section B.3 of the supplementary material. Furthermore, we illustrate our register module in Figure 3. Each vertex in a 3D face mesh is rigged with an embedding, which gets projected to the screen space according to the camera pose, which gives us the 3D feature space on a 2D screen space. These embeddings capture identity-specific information and elicit such information from downstream trainable layers, similar to how registers work in Vision Transformers [3]. We remove the register module during inference, after it has served its purpose of "teaching" LoRA layers to retain fine details.
>
> **Weakness 2 & Question 2: Unsubstantiated Efficiency Claims.**
> We thank the reviewer for their comment. We will incorporate changes in the revised manuscript to highlight the fact that the following two points: (a) The register module is only used during training, as stated in L211-212, and (b) The trained LoRA layers are merged with the original parameters for inference using Eq. 1 in Section 3.2. As a consequence of this implementation, our method retains the exact efficiency of the original model during inference, as does the vanilla LoRA variant.
>
> We provide a table of parameter counts below:
>
>
> | Module Name                      | Parameter Count during Training | Parameter Count during Inference | Inference Time for Variant (FPS) |
> | -------------------------------- | ------------------------------------- | -------------------------- | --------------------------------- |
> | Original Model (w/ DINOv2)       | 199,430,518                           | 199,430,518                 | 34                                |
> | Model + LoRA Layers                      | 204,193,142                             |  199,430,518                 | 34                                |
> | Meta Learning on LoRA            | 204,193,142                            | 199,430,518                  | 34                                |
> | Model + LoRA Layers + 3D Register Module | 222,772,086                            | 199,430,518                 | 34                                |
>
>
> Note that during inference, all variant baselines have the same efficiency, due to the reasons explained above. The FPS numbers are measured based on an Nvidia A6000 GPU.
>
> **Weakness 3: Limited Discussion on Generalizability.**
> We believe that our approach of low-rank personalization is extendable to other approaches because of the use of DINOv2 features, which are general domain features and commonly used in various works. We specifically chose GAGAvatar to demonstrate our method because it is a very competitive method, having been trained on a very large dataset (VFHQ [1]), and producing state-of-the-art results in the general avatar domain.
>
> **Question 3: Temporal Consistency.**
> The video-level prediction is done frame-by-frame. We focus on personalizing GAGAvatar [2] to capture identity-specific details, and thus we do not change its basic architecture. We process the predicted 3DMM parameters using exponential smoothing as in GAGAvatar [2], thereby ensuring temporal consistency in output frames.
>
>
> ## References
> [1] Xie et al., Vfhq: A high-quality dataset and benchmark for video face super-resolution. In CVPR Workshops (CVPRW), 2022.
>
> [2] Chu et al., Generalizable and animatable gaussian head avatar. In NeurIPS 2024.
>
> [3] Darcet et al., ‘Vision Transformers Need Registers’, In ICLR 2024.

---

> > ### Comment · Reviewer_wF51 · 2025-08-07
> >
> > Thank the authors for their detailed and timely rebuttal. I appreciate your clear explanations regarding the architectural details, efficiency claims and generalizability, as well as the supporting evidence provided.
> >
> > Regarding the issue of temporal consistency, while the use of exponential smoothing (as in GAGAvatar) is a common and effective trick in talking head generation, I believe some temporal inconsistencies may still arise in real-world applications if only post-processing is used. It would be worthwhile to explore more fundamental solutions. A brief discussion of this limitation and potential future directions in the revised manuscript would further strengthen the work.
> >
> > Overall, I appreciate the authors’ thorough responses and efforts to address the concerns. I am raising my score to 4.

---

> > > ### Author Response · Authors · 2025-08-08
> > >
> > > Thank you for your suggestion. We would gladly include a discussion regarding temporal consistency in our manuscript. In our work, our goal is to personalize a pre-trained generic model, GAGAvatar, and thus maintain any basic architectural choices of the original generic model. Using their exponential smoothing, we did not observe any noticeable temporal inconsistencies in the outputs.
> > >
> > > However, temporal consistency is crucial in real-world settings and can be further explored as a future work, developing more fundamental solutions. Since 3D head avatar generation methods produce videos in a frame-by-frame manner, using tracked 3DMM parameters, there are two broad directions that can be pursued to make sure temporal consistency is fundamentally “baked-in”: (a) make 3DMM parameter tracking temporally consistent (e.g., predict 3DMM parameters for a window of frames instead of one-by-one [1], etc.) and train and infer the head avatar generation model on these tracked parameters, or (b) make 3D head avatar generation temporally consistent (e.g., predict a window of future frames given a history of frames in a sliding window fashion [2], and/or enable temporal consistency losses using landmark tracking [3], or include other transformer-based modules etc.).
> > >
> > > ## References
> > >
> > > [1] Taubner et al., ‘3D Face Tracking from 2D Video through Iterative Dense UV to Image Flow’, in CVPR, 2024.
> > >
> > > [2] Wu et al., ‘3D-Aware Text-Driven Talking Avatar Generation’, in ECCV 2024.
> > >
> > > [3] Yang et al., ‘Follow Your Motion: A Generic Temporal Consistency Portrait Editing Framework with Trajectory Guidance’, arXiv [cs.CV]. 2025.

---

> > > > ### Author Response · Authors · 2025-08-08
> > > >
> > > > We would like to thank the reviewer once again for their thoughtful and constructive feedback. We will incorporate the additional results, discussion, and clarifications in the revised manuscript, and are happy to address any further questions if needed. We also truly appreciate the reviewer’s earlier note about increasing the score, and are grateful for their time and consideration.

---

### Official Review · Reviewer_xCbz · 2025-07-03

**Clarity:** 3
**Significance:** 2
**Originality:** 3
**Rating:** 4
**Confidence:** 4

**Summary:**

This manuscript proposes a personalized head modeling approach for the domain of personalized avatar modeling, combining low-rank adaptation (LoRA) with a three-dimensional register module. The method aims to enhance the modeling of high-frequency, personalized facial details with minimal parameter updates, effectively addressing the limitations of traditional general-purpose models and existing efficient fine-tuning methods in detail preservation.

**Questions:**

1. The regularization loss $L_{reg}$ in Equation (9) imposes a similarity constraint across all vertex embeddings. While this is effective for preventing overfitting, it may also lead to over-smoothing of features, thereby suppressing the inherent distinctiveness of local regions such as the lips and eyes. The weight for this loss, $\lambda_{reg}$, is set to 20, which is significantly higher than the weight of 2 for the feature alignment loss. Could the authors provide a justification for this choice? I recommend adding an ablation study on the selection of the hyperparameter $\lambda_{reg}$.
2. Following up on question 1, what specific information do the embeddings corresponding to each vertex actually learn? Are there any qualitative conclusions that can be drawn about the learned representations?
3. Could the authors clarify the rationale for summing the `source` features with the features that have been projected by the Register Module? It seems these two sets of features are not aligned to the same viewpoint.
4. I have a question regarding the inference process. The paper states that `$f^{src}_{dense} can be used alone for inference. Could the authors elaborate on the reasoning behind this? Specifically:
- Is the underlying assumption that the LoRA module inherits the capability to render fine-grained details?
- Given that the input to `$f^{src}_{dense} does not seem to contain sufficient high-frequency information, how does the model generate these details effectively?
5. Regarding the quantitative experiments, why were significantly fewer evaluation metrics used compared to those in GAGAvatar?

I will consider raising my score after the authors have addressed my concerns.

**Ethical Concerns:**

["NO or VERY MINOR ethics concerns only"]

**Final Justification:**

This approach is clever and a valuable application of the register module for other fields.

**Limitations:**

Yes

**Paper Formatting Concerns:**

No major formatting issues were found.

**Quality:**

2

**Strengths And Weaknesses:**

### Strengths
- The overall logic of the paper is fairly clear, and the writing is well-structured.
- The core innovation lies in embedding and projecting additional vertex features of the 3D facial mesh through a register module, enabling a deep integration of geometric and visual features. Furthermore, detail interpolation and feature fusion are achieved using a convolutional network.

### Weaknesses
- Its strong dependency on the Register Module's 3DMM parameter accuracy means that errors in pose estimation (from occlusions or extreme angles) cascade, severely degrading the geometric and detailed accuracy of the final image. Another significant drawback is its inability to preserve clothing details, a capability present in earlier keypoint-based methods like FOMM.
- Specific lighting effects from the source image, such as highlights and shadows, are highly prone to being "baked" into the model's texture map or implicit appearance representation. This makes them part of the character's intrinsic identity, which severely limits the model's realism and generalization capability under different lighting conditions.
- During training, the reconstruction branch receives a highly processed feature, $f_{reg}$; however, during inference, it receives a more "raw" DINOv2 feature, $f_{dense}^{src}$. This distribution gap between the training and inference inputs is a potential cause for the model's sub-optimal performance at test time.

---

> ### Author Rebuttal · Authors · 2025-07-31
>
> We thank the reviewer for their thoughtful and constructive feedback. We are encouraged that they find the logic of our paper and idea clear, and recognize that our proposed method effectively addresses the limitations of traditional generalized avatar models and existing fine-tuning methods in detail preservation. We address the reviewer comments below and will incorporate all feedback in the revised manuscript.
>
> **Weakness 1: Dependency on pose estimation; cascading errors.**
> We have not observed any significant degradation for the vast majority of driving frames. This is because we use the state-of-the-art 3DMM parameter estimation algorithms from Inferno:FaceReconstruction [4]. These estimated parameters do well for our purposes. However, the problem the reviewer pointed out might still exist in extreme poses and expressions, as we pointed out in Section 5 of the main paper and Section F.1 in the supplementary paper. Improving pose estimation is an interesting future work. Given the quality of our results, it is out of scope for this paper.
>
> **Clothing details.**
> Cloth modeling is typically considered as a separate modeling problem, independent from face modeling. While FOMM might still be useful for cloth modeling, it is a 2D key-point method that is not as accurate as 3DMM parameters for fine facial motion control. Since our method aims to personalize 3D-aware generic avatar face generation models, cloth modeling is out of our scope. We regret the poor quality of any cloth visualizations; they are presented only for completeness of results. Instead, we point the reviewer to the quality of our results in detailed **face** features such as wrinkles, tattoos, freckles, etc., which is the focus of our research.
>
> **Weakness 2: Bakes source lighting into appearance, harming realism and generalization under new lighting.**
> Our results achieve preservation of specific textural details on faces (such as wrinkles, tattoos, freckles, etc.) in animatable avatars of a generalized avatar generation model. Our work is in line with lighting-agnostic methods (e.g., GAGAvatar[1], GPAvatar [6], Portrait4D [7], etc.) to model avatars. While modeling light for avatar relighting is interesting, it is a separate line of work (e.g., Portrait Video Relighting [8], Holo-Relighting [9]). We leave the task of incorporating lighting effects for the future.
>
> **Weakness 3: Training–inference feature mismatch.**
> As described in L127-132, the Register Module is designed to learn a 3D feature space that captures identity-specific information and elicits such information from downstream trainable layers, similar to how registers work in Vision Transformers [10]. We remove the Register Module during inference, after it has served its purpose of "teaching" LoRA layers to retain fine facial details. This is similar to the way registers were proposed for ViTs [10]. They help store and repurpose information during training and are removed during inference. For completeness, we also tried the case when our Register Module is used during inference; however it harms the output quality - we see an increase in the ACD metric by around 5%.
>
> **Question 1: Regularization Loss for Register Embeddings.**
> Thank you for pointing out a typo in the text. We would like to bring to attention that while equation 9 in the main paper is correct, the text in line 200 is incorrect. As the equation suggests, minimizing the objective $L_{reg} = \frac{\texttt{pcos}(e)}{n(V)(n(V)-1)}$ where $\texttt{pcos}(X) = \sum_i \sum_j \frac{X_i \cdot X_j}{||X_i||||X_j||} - n(V)$ is the sum of the non-diagonal elements of a self-pairwise cosine distance, will result in embeddings $e_i$ to be ***dissimilar*** to each other. We will make a correction in the revised manuscript.
>
> We experimentally found that $\lambda_{reg} = 20$ performs best. We present the ablation study for this hyperparameter below on the VFHQTest dataset.
> | $\lambda_{reg}$ | LPIPS↓ |   ACD↓  |
> |-----------------|--------|--------|
> |     1           | 0.2520 | 0.3616 |
> |     8           | 0.2502 | 0.3606 |
> |    16           | 0.2485 | 0.3594 |
> |    **20**       | **0.2470** | **0.3559** |
> |    32           | 0.2500 | 0.3663 |
>
>
> **Question 2: Qualitative Conclusions on Register Features.**
> As in most current deep-learning methods that use latent spaces, it is impossible to tease out the effect of specific elements on the learnt features. This is the case of Figure 4 in the main paper and Figure 4-11 in the supplementary, where we visualize the projected embeddings. We posit that these features learn information in the 3D space that helps improve learning signals for LoRA, in order to capture fine identity-specific details. Approximating such black-box deep-learning features with networks that use interpretable features, while maintaining accuracy (as in SI-MIL [12]), is interesting future work.
>
> **Question 3 & Question 4: Rationale for Register Pipeline.**
> The Register Module learns a 3D feature space that *stores and repurposes information* for a human face during training, as explained in L127-132. Inspired by registers for ViT [10] that store global information of an image, our Register Module stores the distinctive details of an identity, given different views. We sum the source features with features projected by the Register Module, in order to inject identity-specific information into the representation.
>
> **Question 5: Evaluation Metrics.**
> We found that for our purpose of computing how well the fine details are preserved and the avatar is personalized to the input identity, metrics such as PSNR and SSIM were too unreliable. For example, for the case of the VFHQ Test dataset, we get the following results:
>
> Note: Best and second best results highlighted in **bold** and *italics* respectively.
> | Method        | PSNR↑      | SSIM↑      | LPIPS↓     | ACD↓        |
> |---------------|------------|------------|------------|-------------|
> | GAGAvatar     | **20.86**| *0.5921*   | *0.2540*      | *0.3631* |
> | LoRA          | 20.65    | 0.5699   | 0.2666     | 0.3687  |
> | MetaPortrait  | *20.71*    | **0.5975**| 0.2650  | 0.4131 |
> | Ours          | 20.30    | 0.5423   | **0.2470**| **0.3559** |
>
> MetaPortrait (meta-learning on LoRA weights) gives us the worst performance qualitatively, among all these methods, as can be seen in Figure 6 of the main paper, Figures 2-3 in the supplementary material, and the supplementary video. However, it fares 2nd best in PSNR, and best in SSIM. We believe this is because these metrics do not focus on specific features in faces, but pixel-level features. SSIM specifically is not necessarily designed with human perception in mind (Understanding SSIM [11]). Hence, we rely on the ACD metric and LPIPS, which rely on the perceptual similarity of identity and images, respectively.
>
> ## References
> [1] Chu et al., Generalizable and animatable Gaussian head avatar. In NeurIPS, 2024
>
> [2] Qian et al., Gaussianavatars: Photorealistic head avatars with rigged 3d gaussians. In CVPR, 2024.
>
> [3] Dhamo et al., Headgas: Real-time animatable head avatars via 3d gaussian splatting. In ECCV, 2024.
>
> [4] Daněček et al., Inferno. https://github.com/radekd91/inferno?tab=readme-ov-file
>
> [5] Siarohin et al., ‘First Order Motion Model for Image Animation’, In NeurIPS, 2019.
>
> [6] Chu et al., ‘GPAvatar: Generalizable and Precise Head Avatar from Image(s)’, In ICLR 2024.
>
> [7] Deng,et al., ‘Learning One-Shot 4D Head Avatar Synthesis using Synthetic Data’, In CVPR 2024.
>
> [8] Rao et al., ‘Lite2Relight: 3D-aware Single Image Portrait Relighting’, in Proceedings of ACM SIGGRAPH 2024.
>
> [9] Mei et al., ‘Holo-Relighting: Controllable Volumetric Portrait Relighting from a Single Image’, arXiv preprint arXiv:2403. 09632, 2024.
>
> [10] Darcet, et al., ‘Vision Transformers Need Registers’, in ICLR, 2024.
>
> [11] Nilsson, et al., ‘Understanding SSIM’, arXiv [eess.IV]. 2020.
>
> [12] S. Kapse et al., ‘SI-MIL: Taming Deep MIL for Self-Interpretability in Gigapixel Histopathology’, arXiv preprint arXiv:2312. 15010, 2023.

---

> > ### Comment · Reviewer_xCbz · 2025-08-05
> >
> > Thank you for your rebuttal, which addressed most of my concerns. However, I have questions about the metrics; PSNR might be more suitable for blur, and it's odd that SSIM is low. Overall, I find your approach clever and a valuable application of the register module for other fields.

---

> > > ### Author Response · Authors · 2025-08-06
> > >
> > > We compute the image metrics (PSNR, SSIM and LPIPS) between source image and predicted image on challenging high-frequency feature crops, that include unique facial details of individuals and are the focus of our research. Since there might be a slight misalignment between the generated and ground truth head poses (because of the generation), we find that pixel-based (PSNR) or locality-based (SSIM) metrics to be unreliable. Specifically, PSNR captures blur, but doesn’t capture high-frequency details (which is our main goal), as it still operates on a pixel-by-pixel basis, as in, $MSE = \frac{1}{mn} \sum_{y=0}^{m-1} \sum_{x=0}^{n-1} (i[x,y] - j[x,y])^2$, $PSNR = 10log_{10}\frac{255^2}{MSE}$ where i and j are input images. Moreover, as indicated in the illustration below, and Figure 2 of Understanding SSIM ([1]), we see that while perceptually indistinguishable on high-resolution screens, SSIM fails to recognize the perceptual similarity between the images.
> > > Note that below, we illustrate a 128x128 image using a 4x4 grid. However, the actual SSIM computation was done using the entire 128x128 images (this is only to illustrate the weaknesses of SSIM for perceptually similar images, using a toy example).
> > >
> > > $\text{SSIM}\left(
> > > \begin{bmatrix}
> > > 128 & 128 & 128 & 128 & \cdots \\\\
> > > 128 & 128 & 128 & 128 & \cdots \\\\
> > > 128 & 128 & 128 & 128 & \cdots \\\\
> > > 128 & 128 & 128 & 128 & \cdots \\\\
> > > \vdots & \vdots & \vdots & \vdots & \ddots
> > > \end{bmatrix},
> > > \begin{bmatrix}
> > > 0   & 255 & 0   & 255 & \cdots \\\\
> > > 255 & 0 & 255 & 0 & \cdots \\\\
> > > 0   & 255 & 0   & 255 & \cdots \\\\
> > > 255 & 0 & 255 & 0 & \cdots \\\\
> > > \vdots & \vdots & \vdots & \vdots & \ddots
> > > \end{bmatrix}
> > > \right)
> > > = L (=1) \cdot C (=0.0036) \cdot S (=1) = 0.0036$
> > >
> > > $\text{SSIM}\left(
> > > \begin{bmatrix}
> > > 0   & 255 & 0   & 255 & \cdots \\\\
> > > 255 & 0   & 255 & 0   & \cdots \\\\
> > > 0   & 255 & 0   & 255 & \cdots \\\\
> > > 255 & 0   & 255 & 0   & \cdots \\\\
> > > \vdots & \vdots & \vdots & \vdots & \ddots
> > > \end{bmatrix},
> > > \begin{bmatrix}
> > > 255 & 0   & 255 & 0   & \cdots \\\\
> > > 0   & 255 & 0   & 255 & \cdots \\\\
> > > 255 & 0   & 255 & 0   & \cdots \\\\
> > > 0   & 255 & 0   & 255 & \cdots \\\\
> > > \vdots & \vdots & \vdots & \vdots & \ddots
> > > \end{bmatrix}
> > > \right)
> > > = L (=1) \cdot C (=1) \cdot S (=-0.9964) = -0.9964.$
> > >
> > > Therefore, we rely on LPIPS, as it focuses on perceptual differences between images; the primary task of our research.
> > > ## References
> > > [1] Nilsson, et al., ‘Understanding SSIM’, arXiv [eess.IV]. 2020.

---

> > > > ### Comment · Reviewer_xCbz · 2025-08-07
> > > >
> > > > Thank you for the author's response. I will increase my rating accordingly.

---

> > > > > ### Author Response · Authors · 2025-08-08
> > > > >
> > > > > Thank you for your thoughtful review, constructive feedback, and helpful suggestions. We truly appreciate your time and are grateful that you will be raising the score.

---

### Author Response · Authors · 2025-08-05
**Invitation for Discussion**

We sincerely thank the reviewers for their thoughtful and encouraging feedback. We are grateful for their recognition of the clarity and structure of our paper, the novelty and effectiveness of the Register Module in enhancing high-fidelity personalization beyond standard LoRA, and the value of our RareFace-50 dataset in addressing underrepresented facial features. We have submitted our rebuttal addressing all raised points and would like to kindly ask if there are any remaining questions or concerns.

---

### Note · Authors · 2025-08-13

We thank the reviewers for their thoughtful and constructive feedback.

**Reviewer-Identified Strengths**

We present the strengths of our paper as identified by the reviewers below:
* They found our work clear, well-structured, and significant, addressing the problem of high-fidelity personalization of generic 3D-aware head avatar models.
* They appreciated the novelty and simplicity of our 3D Register Module to capture high-frequency identity-specific details.
* They valued our strong results over LoRA, meta-learning, and other methods.
* They also noted the contribution of the RareFace-50 dataset for distinctive and underrepresented attributes.

**Reviewer Suggestions**

Reviewers raised helpful suggestions during rebuttal and subsequent discussion, including:
* expand comparisons to DoRA and single-identity models (e.g., SplattingAvatar) (Reviewer QQ5r);
* clarify evaluation metrics (Reviewer xCbz);
* clarify architectural details and efficiency claims (Reviewer wF51, xCbz);
* include additional evaluation on a 3rd dataset (Reviewer k9TX);
* adding additional ablations (loss weights, noisy/xyz vertex embeddings) (Reviewer xCbz, QQ5r).

**How we addressed reviewer suggestions**

We have addressed the above suggestions, demonstrating additional results and explaining any clarifications needed. In summary, we have added the following changes in the revised manuscript:
* Included the additional ablations and results as described in our replies.
* Included additional discussion on limitations and potential future work regarding pose, lighting, and temporal consistency.
* Included additional citations.
* Clarified distinction from LAM and single-identity head reconstruction.

All the aforementioned additions have been described in detail in our replies. The reviewers have not raised any additional questions.

**Note to AC**: We would like to bring to your attention that **Reviewer wF51 indicated to raise the score to 4** in the follow-up comment, but the score has not yet been updated in the system. We sincerely hope that the reviewer will raise the score by the end of the discussion period.

We would again like to thank the reviewers and AC for their time, effort, and valuable feedback.

---

### Decision · Program_Chairs · 2025-09-17

**Decision:**

Accept (poster)

**Comment:**

The paper studies head avatar generation, and proposes a register module that enhances the performance of LoRA adapters, while requiring only a small number of parameters to adapt to an unseen identity. Initial reviews raised several issues, including clarifications of the contributions, limited empirical evidence, and related work comparisons. The rebuttal successfully addressed most of these concerns, leading to two reviewers raising their scores. After careful deliberation, the decision was made to accept this paper - congratulations to the authors! It is crucial, however, that all provided clarifications are incorporated in the final version of the paper.